# Cooperative Online Learning with Feedback Graphs

**Nicolò Cesa-Bianchi**                                            *nicolo.cesa-bianchi@unimi.it*
*Politecnico di Milano & Università degli Studi di Milano, Milano, Italy*

**Tommaso Cesari**                                                        *tcesari@uottawa.ca*
*School of Electrical Engineering and Computer Science, University of Ottawa, Ottawa, Canada*

**Riccardo Della Vecchia**                                      *ric.della.vecchia@gmail.com*
*Inria, Université de Lille, CNRS, Centrale Lille, UMR 9189 – CRIStAL*

**Reviewed on OpenReview:** *https: // openreview. net/ forum? id= PtNyIboDIG*

## Abstract

We study the interplay between communication and feedback in a cooperative online learning setting, where a network of communicating agents learn a common sequential decision-making task through a feedback graph. We bound the network regret in terms of the independence number of the strong product between the communication network and the feedback graph. Our analysis recovers as special cases many previously known bounds for cooperative online learning with expert or bandit feedback. We also prove an instance-based lower bound, demonstrating that our positive results are not improvable except in pathological cases. Experiments on synthetic data confirm our theoretical findings.

## 1 Introduction

Nonstochastic online learning with feedback graphs (Mannor and Shamir, 2011) is a sequential decision-making setting in which, at each decision round, an oblivious adversary assigns losses to all actions in a finite set. What the learner observes after choosing an action is determined by a feedback graph defined on the action set. Unlike bandit feedback, where a learner choosing an action pays and observes the corresponding loss, in the feedback graph setting the learner also observes (without paying) the loss of all neighboring actions in the graph. Special cases of this setting are prediction with expert advice (where the graph is a clique) and multiarmed bandits (where the graph has no edges). The Exp3-SET algorithm (Alon et al., 2017) is known to achieve a regret scaling with the square root of the graph independence number, and this is optimal up to a logarithmic factor in the number of actions. In recommendation systems, feedback graphs capture situations in which a user's reaction to a recommended product allows the system to infer what reaction similar recommendations would have elicited in the same user, see Alon et al. (2017) for more examples.

Online learning has been also investigated in distributed settings, in which a network of cooperating agents solves a common task. At each time step, some agents become active, implying that they are requested to make predictions and pay the corresponding loss. Agents cooperate through a communication network by sharing the feedback obtained by the active agents. The time this information takes to travel the network is taken into account: a message broadcast by an agent is received by another agent after a delay equal to the shortest path between them. Regret in cooperative online learning has been previously investigated only in the full-information setting (Cesa-Bianchi et al., 2020; Hsieh et al., 2022) and in the bandit setting (Cesa-Bianchi et al., 2019; Bar-On and Mansour, 2019a).

In this work we provide a general solution to the problem of cooperative online learning with feedback graphs. In doing so, we generalize previous approaches and also clarify the impact on the regret of the mechanism governing the activation of agents. Under the assumption that agents are stochastically activated, our analysis captures the interplay between the communication graph (over the agents) and the feedback graph (over

the actions), showing that the network regret scales with the independence number of the strong product between the communication network and the feedback graph.

More precisely, we design a distributed algorithm, $\textsc{Exp3-}\alpha^2$, whose average regret $R_T/Q$ (where $Q$ is the expected number of simultaneously active agents) on any communication network $N$ and any feedback graph $F$ is (up to log factors)

$$\frac{R_T}{Q} \overset{\mathcal{O}}{=} \sqrt{\left(\frac{\alpha\left(N^n \boxtimes F\right)}{Q} + 1 + n\right) T}\,, \tag{1}$$

where $T$ is the horizon, $n$ is the diffusion radius (the maximum delay after which feedback is ignored), and $\alpha\left(N^n \boxtimes F\right)$ is the independence number of the strong product between the $n$-th power $N^n$ of the communication network $N$ and the feedback graph $F$. We also prove a near-matching instance-dependent lower bound showing that, with the exception of pathological cases, for any pair of graphs $(N, F)$, no algorithm run with an oblivious network interface can have regret smaller than $\sqrt{(\alpha(N^n \boxtimes F)/Q)T}$.

Our results hold for any diffusion radius $n$, which serves as a parameter to control the message complexity of the protocol. When $n$ is equal to the diameter of the communication network, then every agent can communicate with every other agent. Our protocol is reminiscent of the $\textsc{local}$ communication model in distributed computing (Linial, 1992; Suomela, 2013), where the output of a node depends only on the inputs of other nodes in a constant-size neighborhood of it, and the goal is to derive algorithms whose running time is independent of the network size. Although our tasks have no completion time, in our model each node is directly influenced only by a constant-size neighborhood around it.

Let $|A|$ and $|K|$ be, respectively, the number of agents and actions. When $Q = |A|$ (all agents are always active) and $F$ is the bandit graph (no edges), then $\alpha(N^n \boxtimes F) = |K|\alpha(N^n)$ and we recover the bound $\sqrt{(\alpha(N^n)|K|/|A| + 1 + n)T}$ of Cesa-Bianchi et al. (2019). When $n = 1$ and $F$ is the expert graph (clique), then $\alpha(N^n \boxtimes F) = \alpha(N)$ and we recover the bound $\sqrt{(\alpha(N)/Q + 1)T}$ of Cesa-Bianchi et al. (2020)[1]. Interestingly, if all agents were always active in Cesa-Bianchi et al. (2020), the graph topology would become irrelevant in the expert setting, resulting in a simplified regret bound of $\mathcal{O}(\sqrt{T})$, analogous to the case of a clique graph. This starkly contrasts with the bandit case of Cesa-Bianchi et al. (2019), where even when all agents are active simultaneously, the graph topology explicitly appears in the regret bound. Finally, in the non-cooperative case ($N$ is the bandit graph), we obtain $\sqrt{|A|\alpha(F)T/Q}$ which, for $|A| = 1$ and $Q = 1$, recovers the bound of Alon et al. (2017). Table 1 summarizes all known bounds (omitting log factors and setting, for simplicity, $Q = 1$ and $n = 0$).

| | $|N| = 1$ | | Any $N$ | |
|---|---|---|---|---|
| $F$ = clique (experts) | $\sqrt{T}$ | (Freund and Schapire, 1997) | $\sqrt{\alpha(N)T}$ | (Cesa-Bianchi et al., 2020) |
| $F$ = no edges (bandits) | $\sqrt{|K|T}$ | (Auer et al., 2002) | $\sqrt{\alpha(N)|K|T}$ | (Cesa-Bianchi et al., 2019) |
| Any $F$ | $\sqrt{\alpha(F)T}$ | (Alon et al., 2017) | $\sqrt{\alpha(N \boxtimes F)T}$ | (this work) |

Table 1: Known bounds in online learning with feedback graphs and cooperative online learning.

Our theory is developed under the so-called oblivious network interface; i.e., when agents are oblivious to the global network topology and run an instance of the same algorithm using a common initialization and a common learning rate for their updates. In this case, the stochastic activation assumption is necessary to not incur linear regret $R_T = \Omega(T)$ Cesa-Bianchi et al. (2020).

Our core and main technical contributions are presented in Lemma 1 and Theorem 1 for the upper bound, and in Lemma 2 and Theorem 2 for the lower bound. Lemma 1, implies that the second moment of the loss estimates is dominated by the independence number of the strong product between the two graphs. The proof of this result generalizes the analysis of (Cesa-Bianchi et al., 2019, Lemma 3), identifying the strong product

---

[1]This is a reformulation of the bound originally proven by Cesa-Bianchi et al. (2020), see Section C of the Supplementary Material for a proof.

as the appropriate notion for capturing the combined effects of the communication and feedback graphs. In Theorem 1, we present a new analysis, in the distributed learning setting, of the "drift" term arising from the decomposition of network regret. This is obtained by combining Lemma 1 with a regret analysis technique developed by Gyorgy and Joulani (2021) for a single agent. The proof of the lower bound in Theorem 2 builds upon a new reduction to the setting of Lemma 2 that we prove in Appendix A. Lemma 2 contains a lower bound for a single-agent setting with a feedback graph and oblivious adversary where every time step is independently skipped with a known and constant probability $q$. This reduction is new and necessary, since it is not enough to claim that the average number of rounds played is $qT$ and plug this in the lower bound for bandit with feedback graphs. In fact, one needs to build an explicit assignment of $\ell_1, \ldots, \ell_T$ such that by averaging over the random subset of active time steps it is possible to prove the lower bound under the conditions detailed in Lemma 2. In Section 6, we corroborate our theoretical results with experiments on synthetic data.

## 2    Further related work

**Adversarial losses.**    A setting closely related to ours is investigated by Herbster et al. (2021). However, they assume that the learner has full knowledge of the communication network—a weighted undirected graph—and provide bounds for a harder notion of regret defined with respect to an unknown smooth function mapping users to actions. Bar-On and Mansour (2019b) bound the individual regret (as opposed to our network regret) in the adversarial bandit setting of Cesa-Bianchi et al. (2019), in which all agents are active at all time steps. Their results, as well as the results of Cesa-Bianchi et al. (2019), have been extended to cooperative linear bandits by Ito et al. (2020). Della Vecchia and Cesari (2021) study cooperative linear semibandits and focus on computational efficiency. Dubey et al. (2020a) show regret bounds for cooperative contextual bandits, where the reward obtained by an agent is a linear function of the contexts. Nakamura et al. (2023) consider cooperative bandits in which agents dynamically join and leave the system.

**Stochastic losses.**    Cooperative stochastic bandits are also an important topic in the online learning community. Kolla et al. (2018) study a setting in which all agents are active at all time steps. In our model, this corresponds to the special case where the feedback graph is a bandit graph (no edges) and the activation probabilities $q(v)$ are equal to 1 for all agents $v$. More importantly, however, they focus on a stochastic multi-armed bandit problem. Hence, even restricting to the special cases of bandits with simultaneous activation, their algorithmic ideas cannot be directly applied to our adversarial setting. Other recently studied variants of cooperative stochastic bandits consider agent-specific restrictions on feedback (Chen et al., 2021) or on access to arms (Yang et al., 2022), bounded communication (Martínez-Rubio et al., 2019), corrupted communication (Madhushani et al., 2021), heavy-tailed reward distributions (Dubey et al., 2020b), stochastic cooperation models (Chawla et al., 2020), strategic agents (Dubey and Pentland, 2020), and Bayesian agents (Lalitha and Goldsmith, 2021). Multi-agent bandits have been also studied in federated learning settings with star-shaped communication networks (He et al., 2022; Li and Wang, 2022) in the presence of adversarial (as opposed to stochastic) agent activations. Finally, Liu et al. (2021) investigate a decentralized stochastic bandit network for matching markets.

## 3    Notation and setting

Our graphs are undirected and contain all self-loops. For any undirected graph $G = (V, E)$ and all $m \geq 0$, we let $\delta_G(u, v)$ be the *shortest-path distance* (in $G$) between two vertices $u, v \in V$, $G^m$ the $m$-th power of $G$ (i.e., the graph with the same set of vertices $V$ of $G$ but in which two vertices $u, v \in V$ are adjacent if and only if $\delta_G(u, v) \leq m$), $\alpha(G)$ the *independence number* of $G$ (i.e., the largest cardinality of a subset $I$ of $V$ such that $\delta_G(u, v) > 1$ for all distinct $u, v \in I$), and $\mathcal{N}^G(v)$ the *neighborhood* $\{u \in V : \delta_G(u, v) \leq 1\}$ of a vertex $v \in V$. To improve readability, we sometimes use the alternative notations $\alpha_m(G)$ for $\alpha(G^m)$ and $\mathcal{N}_m^G(v)$ for $\mathcal{N}^{G^m}(v)$. Finally, for any two undirected graphs $G = (V, E)$ and $G' = (V', E')$, we denote by $G \boxtimes G'$ their *strong product*, defined as the graph with set of vertices $V \times V'$ in which $(v, v')$ is adjacent to $(u, u')$ if and only if $(v, v') \in \mathcal{N}^G(u) \times \mathcal{N}^{G'}(u')$.

An instance of our problem is parameterized by:

1. A **communication network** $N = (A, E_N)$ over a set $A$ of agents, and a maximum communication delay $n \geq 0$, limiting the communication among agents.
2. A **feedback graph** $F = (K, E_F)$ over a set $K$ of actions.
3. An **activation probability** $q(v) > 0$ for each agent $v \in A$,[2] determining the subset of agents incurring losses on that round. Let $Q = \sum_{v \in A} q(v)$ be the expected cardinality of this subset.
4. A sequence $\ell_1, \ell_2, \ldots : K \to [0, 1]$ of **losses**, chosen by an oblivious adversary.

We assume the agents do not know $N$ (see the oblivious network interface assumption introduced later). The only assumption we make is that each agent knows the pairs $\big(v, q(v)\big)$ for all agents $v$ located at distance $n$ or less.[3]

The distributed learning protocol works as follows. At each round $t = 1, 2, \ldots$, each agent $v$ is activated with a probability $q(v)$, independently of the past and of the other agents. Agents that are not activated at time $t$ remain inactive for that round. Let $\mathcal{A}_t$ be the subset of agents that are activated at time $t$. Each $v \in \mathcal{A}_t$ plays an action $I_t(v)$ drawn according to its current probability distribution $p_t(\cdot, v)$, is charged the corresponding loss $\ell_t\big(I_t(v)\big)$, and then observes the losses $\ell_t(i)$, for any action $i \in \mathcal{N}_1^F\big(I_t(v)\big)$. Afterwards, each agent $v \in A$ broadcasts to all agents $u \in \mathcal{N}_n^N(v)$ in its $n$-neighborhood a feedback message containing all the losses observed by $v$ at time $t$ together with its current distribution $p_t(\cdot, v)$; any agent $u \in \mathcal{N}_n^N(v)$ receives this message at the end of round $t + \delta_N(v, u)$. Note that broadcasting a message in the $n$-neighborhood of an agent $v$ can be done when $v$ knows only its 1-neighborhood. Indeed, because messages are time-stamped using a global clock, $v$ drops any message received from an agent outside its $n$-neighborhood. On the other hand, $v$ may receive more than once the same message sent by some agent in its $n$-neighborhood. To avoid making a double update, an agent can extract from each received message the timestamp together with the index of the sender, and keep these pairs stored for $n$ time steps.

Each loss observed by an agent $v$ (either directly or in a feedback message) is used to update its local distribution $p_t(\cdot, v)$. To simplify the analysis, updates are postponed, i.e., updates made at time $t$ involve only losses generated at time $t - n - 1$. This means that agents may have to store feedback messages for up to $n + 1$ time steps before using them to perform updates.

The online protocol can be written as follows.

---

At each round $t = 1, 2, \ldots$
1. Each agent $v$ is independently activated with probability $q(v)$;
2. Each active agent $v$ draws an action $I_t(v)$ from $K$ according to its current distribution $p_t(\cdot, v)$, is charged the corresponding loss $\ell_t\big(I_t(v)\big)$, and observes the set of losses $\mathcal{L}_t(v) = \big\{\big(i, \ell_t(i)\big) : i \in \mathcal{N}_1^F\big(I_t(v)\big)\big\}$
3. Each agent $v$ broadcasts to all agents $u \in \mathcal{N}_n^N(v)$ the feedback message $\big(t, v, \mathcal{L}_t(v), p_t(\cdot, v)\big)$, where $\mathcal{L}_t(v) = \varnothing$ if $v \notin \mathcal{A}_t$
4. Each agent $v$ receives the feedback message $\big(t - s, u, \mathcal{L}_{t-s}(u), p_{t-s}(\cdot, u)\big)$ from each agent $u$ such that $\delta_N(v, u) = s$, for all $s \in [n]$

---

Similarly to Cesa-Bianchi et al. (2019), we assume the feedback message sent out by an agent $v$ at time $t$ contains the distribution $p_t(\cdot, v)$ used by the agent to draw actions at time $t$. This is needed to compute the importance-weighted estimates of the losses, $b_t(i, v)$, see (3).

The goal is to minimize the *network regret*

$$R_T = \max_{i \in K} \mathbb{E}\left[\sum_{t=1}^T \sum_{v \in \mathcal{A}_t} \ell_t\big(I_t(v)\big) - \sum_{t=1}^T |\mathcal{A}_t|\, \ell_t(i)\right], \tag{2}$$

---

[2]We assume without loss of generality that $q(v) \neq 0$ for all agents $v \in A$. The definition of regret (2) and all subsequent results could be restated equivalently in terms of the restriction $N' = (A', E'_N)$ of the communication network $N$, where $A' = \{v \in A : q(v) > 0\}$ and for all $u, v \in A'$, $(u, v) \in E'_N$ if and only if $(u, v) \in E'_N$.

[3]This assumption can be relaxed straightforwardly by assuming that each agent $v$ only knows $q(v)$, which can then be broadcast to the $n$-neighborhood of $v$ as the process unfolds.

where the expectation is taken with respect to the activations of the agents and the internal randomization of the strategies drawing the actions $I_t(v)$. Since the active agents $\mathcal{A}_t$ are chosen i.i.d. from a fixed distribution, we also consider the average regret

$$\frac{R_T}{Q} = \frac{1}{Q}\mathbb{E}\left[\sum_{t=1}^{T}\sum_{v\in\mathcal{A}_t}\ell_t\big(I_t(v)\big)\right] - \min_{i\in K}\sum_{t=1}^{T}\ell_t(i)\ ,$$

where $Q = \mathbb{E}\big[|\mathcal{A}_t|\big] > 0$ for all $t$.

In our setting, each agent locally runs an instance of the same online algorithm. We do not require any ad-hoc interface between each local instance and the rest of the network. In particular, we make the following assumption (Cesa-Bianchi et al., 2020).

**Assumption 1** (Oblivious network interface)**.** *An online algorithm* ALG *is run with an* oblivious network interface *if:*

1. *each agent $v$ locally runs a local instance of* ALG*;*
2. *all local instances use the same initialization and the same strategy for updating the learning rates;*
3. *all local instances make updates while being oblivious to whether or not their host node $v$ was active and when.*

This assumption implies that each agent's instance is oblivious to both the network topology and the location of the agent in the network. Its purpose is to show that communication improves learning rates even without any network-specific tuning. In concrete applications, one might use ad-hoc variants that rely on the knowledge of the task at hand, and decrease the regret even further.

## 4   Upper bound

In this section, we introduce EXP3-$\alpha^2$ (Algorithm 1), an extension of the EXP3-COOP algorithm by Cesa-Bianchi et al. (2019), and analyze its network regret when run with an oblivious network interface.

---
**Algorithm 1:**   EXP3-$\alpha^2$ (Locally run by each agent $v \in A$)

---
**input:** learning rates $\eta_1(v), \eta_2(v)\dots$
**for** $t = 1, 2, \dots, n+1$ **do**
    if $v$ is active in this round, draw $I_t(v)$ from $K$ uniformly at random
**for** $t \geq n+2$ **do**
    if $v$ is active in this round, draw $I_t(v)$ from $K$ according to $p_t(\cdot, v)$ in (3)

---

An instance of EXP3-$\alpha^2$ is locally run by each agent $v \in A$. The algorithm is parameterized by its (variable) learning rates $\eta_1(v), \eta_2(v), \dots$, which, in principle, can be arbitrary (measurable) functions of the history. In all rounds $t$ in which the agent is active, $v$ draws an action $I_t(v)$ according to a distribution $p_t(\cdot, v)$. For the first $n+1$ rounds $t$, $p_t(\cdot, v)$ is the uniform distribution over $K$. During all remaining time steps $t$, the algorithm computes exponential weights using all the available feedback generated up to (and including) round $t - n - 1$. More precisely, for any action $i \in K$,

$$
\begin{aligned}
p_t(i,v) &= w_t(i,v)/\left\|w_t(\cdot,v)\right\|_1\ , \\
w_t(i,v) &= \exp\big(-\eta_t(v)\textstyle\sum_{s=1}^{t-n-1}\widehat{\ell}_s(i,v)\big)\ , \\
\widehat{\ell}_s(i,v) &= \ell_s(i)B_s(i,v)/b_s(i,v)\ , \\
B_s(i,v) &= \mathbb{I}\big\{\exists u \in \mathcal{N}_n^N(v) : u \in \mathcal{A}_s, I_s(u) \in \mathcal{N}_1^F(i)\big\}\ , \\
b_s(i,v) &= 1 - \textstyle\prod_{u\in\mathcal{N}_n^N(v)}\big(1 - q(u)\sum_{j\in\mathcal{N}_1^F(i)}p_s(j,u)\big)\ .
\end{aligned}
\tag{3}
$$

The event $B_s(i,v)$ indicates whether an agent in the $n$-neighborhood of $v$ played at time $s$ an action in the 1-neighborhood of $i$. If $B_s(i,v)$ occurs, then agent $v$ can use $\widehat{\ell}_s(i,v)$ to update the local estimate for the

cumulative loss of $i$. Note that $\widehat{\ell}_s(i,v)$ are proper importance-weighted estimates, as $\mathbb{E}_{s-n}\big[\widehat{\ell}_s(i,v)\big] = \ell_s(i)$ for all $v \in A$, $i \in K$, and $s > n$. The notation $\mathbb{E}_{s-n}$ denotes conditioning with respect to any randomness in rounds $1, \ldots, s-n-1$. Note also that when $q(u) = 1$ for all $u \in A$ and $F$ is the edgeless graph, the probabilities $p_t(i,v)$ in (3) correspond to those computed by Exp3-Coop (Cesa-Bianchi et al., 2019).

Before analyzing our cooperative implementation of Exp3-$\alpha^2$, we present a key graph-theoretic result that helps us characterize the joint impact on the regret of the communication network and the feedback graph. Our new result relates the variance of the estimates of eq. (3) to the structure of the communication graph given by the strong product of $N^n$ and $F$.

**Lemma 1.** *Let $N = (A, E_N)$ and $F = (K, E_F)$ be any two graphs, $n \geq 0$, $\big(q(v)\big)_{v \in A}$ a set of numbers in $(0,1]$, $Q = \sum_{v \in A} q(v)$, and $\big(p(i,v)\big)_{i \in K, v \in A}$ a set of numbers in $(0,1]$ such that $\sum_{i \in K} p(i,v) = 1$ for all $v \in A$. Then,*

$$\sum_{v \in A} \sum_{i \in K} \frac{q(v)p(i,v)}{1 - \prod_{u \in \mathcal{N}_n^N(v)} \big(1 - q(u) \sum_{j \in \mathcal{N}_1^F(i)} p(j,u)\big)} \leq \frac{1}{1 - e^{-1}} \left( \alpha\big(N^n \boxtimes F\big) + Q \right) .$$

*Proof.* Let $\boldsymbol{w} = \big(w(i,v)\big)_{(i,v) \in K \times A}$ where $w(i,v) = q(v)p(i,v)$, and for all $(i,v) \in K \times A$ set $W(i,v) = \sum_{(j,u) \in \mathcal{N}_1^F(i) \times \mathcal{N}_n^N(v)} w(j,u)$. Define also, for all $\boldsymbol{c} = \big(c(j,u)\big)_{(j,u) \in K \times A} \in [0,1]^{|K| \times |A|}$ and $(i,v) \in K \times A$

$$f_{\boldsymbol{c}}(i,v) = 1 - \prod_{u \in \mathcal{N}_n^N(v)} \left( 1 - \sum_{j \in \mathcal{N}_1^F(i)} c(j,u) \right) .$$

Then we can write the left-hand side of the statement of the lemma as

$$\sum_{(i,v) \in K \times A} \frac{w(i,v)}{f_{\boldsymbol{w}}(i,v)} = \underbrace{\sum_{(i,v) \in K \times A : W(i,v) < 1} \frac{w(i,v)}{f_{\boldsymbol{w}}(i,v)}}_{(I)} + \underbrace{\sum_{(i,v) \in K \times A : W(i,v) \geq 1} \frac{w(i,v)}{f_{\boldsymbol{w}}(i,v)}}_{(II)}$$

and proceed by upper bounding the two terms (I) and (II) separately. For the first term (I), using the inequality $1 - x \leq e^{-x}$ (for all $x \in \mathbb{R}$) with $x = w(j,u)$, we can write, for any $(i,v) \in K \times A$,

$$f_{\boldsymbol{w}}(i,v) \geq 1 - \exp\left( -\sum_{u \in \mathcal{N}_n^N(v)} \sum_{j \in \mathcal{N}_1^F(i)} w(j,u) \right) = 1 - \exp\big(-W(i,v)\big) .$$

Now, since in (I) we are only summing over $(i,v) \in K \times A$ such that $W(i,v) < 1$, we can use the inequality $1 - e^{-x} \geq (1 - e^{-1})x$ (for all $x \in [0,1]$) with $x = W(i,v)$, obtaining $f_{\boldsymbol{w}}(i,v) \geq (1 - e^{-1})W(i,v)$, and in turn

$$(I) \leq \sum_{(i,v) \in K \times A : W(i,v) < 1} \frac{w(i,v)}{(1 - e^{-1})W(i,v)} \leq \frac{1}{1 - e^{-1}} \sum_{(i,v) \in K \times A} \frac{w(i,v)}{W(i,v)} \leq \frac{\alpha\big(N^n \boxtimes F\big)}{1 - e^{-1}} ,$$

where in the last step we used a known graph-theoretic result—see Lemma 3 in the Supplementary Material. For the second term (II): for all $v \in A$, let $r(v)$ be the cardinality of $\mathcal{N}_n^N(v)$. Then, for any $(i,v) \in K \times A$ such that $W(i,v) \geq 1$,

$$1 - f_{\boldsymbol{w}}(i,v) \leq \max\left\{ 1 - f_{\boldsymbol{c}}(i,v) : \boldsymbol{c} \in [0,1]^{|K| \times |A|}, \sum_{(j,u) \in \mathcal{N}_1^F(i) \times \mathcal{N}_n^N(v)} c(j,u) \geq 1 \right\}$$

$$= \max\left\{ \prod_{u \in \mathcal{N}_n^N(v)} \left( 1 - \sum_{j \in \mathcal{N}_1^F(i)} c(j,u) \right) : \boldsymbol{c} \in [0,1]^{|K| \times |A|}, \sum_{u \in \mathcal{N}_n^N(v)} \sum_{j \in \mathcal{N}_1^F(i)} c(j,u) = 1 \right\}$$

$$\leq \max\left\{ \prod_{u \in \mathcal{N}_n^N(v)} \big( 1 - C(u) \big) : \boldsymbol{C} \in [0,1]^{|A|}, \sum_{u \in \mathcal{N}_n^N(v)} C(u) = 1 \right\}$$

$$= \max\left\{ \prod_{u\in\mathcal{N}_n^N(v)}\big(1-C(u)\big) \ : \ \boldsymbol{C}\in[0,1]^{|A|}, \ \sum_{u\in\mathcal{N}_n^N(v)}\big(1-C(u)\big) = r(v)-1 \right\}$$

$$\le \left(1-\frac{1}{r(v)}\right)^{r(v)} \le e^{-1} \ ,$$

where the first equality follows from the definition of $f_{\boldsymbol{c}}(i,v)$ and the monotonicity of $x\mapsto 1-x$, the second-to-last inequality is implied by the AM-GM inequality (Lemma 4), and the last one comes from $r(v)\ge 1$ (being $v\in\mathcal{N}_n^N(v)$). Hence

$$(\mathrm{II}) = \sum_{(i,v)\in K\times A\,:\,W(i,v)\ge 1}\frac{w(i,v)}{f_{\boldsymbol{w}}(i,v)} \le \sum_{(i,v)\in K\times A\,:\,W(i,v)\ge 1}\frac{w(i,v)}{1-e^{-1}}$$

$$\le \frac{1}{1-e^{-1}}\sum_{i\in K}\sum_{v\in A}w(i,v) = \frac{1}{1-e^{-1}}\sum_{v\in A}q(v)\sum_{i\in K}p(i,v) = \frac{Q}{1-e^{-1}} \ .$$

$\square$

For a slightly stronger version of this result, see Lemma 6 in the Supplementary Material. By virtue of Lemma 1, we can now show the main result of this section.

**Theorem 1.** *If each agent $v\in A$ uses adaptive learning rates equal to $\eta_t(v) = 0$ for $t\le n+1$, $\eta_t(v) = \sqrt{\log(K)/\sum_{s=1}^t X_s(v)}$ with $X_t(v) = n + \sum_{i\in K}\frac{p_t(i,v)}{b_t(i,v)}$ for $t > n+1$, the average network regret of $\mathrm{EXP3}\text{-}\alpha^2$ playing with an oblivious network interface can be bounded as*

$$\frac{R_T}{Q} \stackrel{\widetilde{\mathcal{O}}}{=} \sqrt{\log(K)\left(n+1+\frac{\alpha\left(N^n\boxtimes F\right)}{Q}\right)T} \ . \tag{4}$$

*Proof.* Let $i^\star\in\arg\min_{i\in K}\mathbb{E}\big[\sum_{t=1}^T |\mathcal{A}_t|\,\ell_t(i)\big]$ where the expectation is with respect to the random sequence $\mathcal{A}_t\subseteq A$ of agent activations. We write the network regret $R_T$ as a weighted sum of single agent regrets $R_T(v)$:

$$R_T = \sum_{v\in A}q(v)R_T(v) = \sum_{v\in A}q(v)\mathbb{E}\left[\sum_{t=1}^T\sum_{i\in K}\widehat{\ell}_t(i,v)p_t(i,v) - \widehat{\ell}_t(i^\star,v)\right] \ ,$$

where the expectation is now only with respect to the random draw of the agents' actions, and it is separated from the activation probability $q(v)$. Fix any agent $v\in A$. $\mathrm{EXP3}\text{-}\alpha^2$ plays uniformly for the first $n+1$ rounds, and each agent, therefore, incurs a linear regret in this phase. For $t > n+1$ we borrow a decomposition technique from (Gyorgy and Joulani, 2021): for any sequence $\big(\widetilde{p}_t(\cdot,v)\big)_{t>n+1}$ of distributions over $K$, the above expectation can be written as

$$\mathbb{E}\left[\sum_{t=n+2}^T\sum_{i\in K}\widehat{\ell}_t(i,v)\widetilde{p}_{t+1}(i,v) - \widehat{\ell}_t(i^\star,v)\right] + \sum_{t=n+2}^T\mathbb{E}\left[\sum_{i\in K}\widehat{\ell}_t(i,v)p_t(i,v)\left(1-\frac{\widetilde{p}_{t+1}(i,v)}{p_t(i,v)}\right)\right] \ . \tag{5}$$

Take now $\widetilde{p}_t(\cdot,v)$ as the (full-information) exponential-weights updates with non-increasing step-sizes $\eta_{t-1}(v)$ for the sequence of losses $\widehat{\ell}_t(\cdot,v)$. That is, $\widetilde{p}_1(\cdot,v)$ is the uniform distribution over $K$, and for any time step $t$ and action $i\in K$, $\widetilde{p}_{t+1}(i,v) = \widetilde{w}_{t+1}(i,v)/\|\widetilde{w}_{t+1}(\cdot,v)\|_1$, where $\widetilde{w}_{t+1}(i,v) = \exp\big(-\eta_t(v)\sum_{s=1}^t\widehat{\ell}_s(i,v)\big)$. With this choice, the first term in (5) is the "look-ahead" regret for the iterates $\widetilde{p}_{t+1}(\cdot,v)$ (which depend on $\widehat{\ell}_t(\cdot,v)$ at time $t$), while the second one measures the drift of $p_t(\cdot,v)$ from $\widetilde{p}_{t+1}(\cdot,v)$.

Using an argument from (Joulani et al., 2020, Theorem 3),[4] we deterministically bound the first term in (5):

$$\sum_{t=n+2}^T\sum_{i\in K}\widehat{\ell}_t(i,v)\widetilde{p}_{t+1}(i,v) - \widehat{\ell}_t(i^\star,v) \le \frac{\ln|K|}{\eta_T(v)} \ . \tag{6}$$

---

[4]We use (Joulani et al., 2020, Theorem 3) with $p_t = 0$ for all $t\in[T]$, $r_0 = (1/\eta_0(v))\sum_i p_i\ln(p_i)$, $r_t(p) = (1/\eta_t(v) - 1/\eta_{t-1}(v))\sum_i p_i\ln(p_i)$ for all $t\in[T]$, and dropping the Bregman-divergence terms due to the convexity of $r_t$.

The subtle part is now to control the second term in (5). To do so, fix any $t > n + 1$. Note that for all $i \in K$,

$$w_t(i, v) = \exp\left(-\eta_t(v) \sum_{s=1}^{t-n-1} \widehat{\ell}_s(i, v)\right) \geq \exp\left(-\eta_t(v) \sum_{s=1}^{t} \widehat{\ell}_s(i, v)\right) = \widetilde{w}_{t+1}(i, v)$$

(using $\ell_s(i, v) \geq 0$ for all $s, i, v$), which in turn, using the inequality $e^x \geq 1 + x$ (for all $x \in \mathbb{R}$), yields

$$\frac{\widetilde{p}_{t+1}(i, v)}{p_t(i, v)} \geq \frac{\widetilde{w}_{t+1}(i, v)}{w_t(i, v)} = \exp\left(-\eta_t(v) \textstyle\sum_{s=t-n}^{t} \widehat{\ell}_s(i, v)\right) \geq 1 - \eta_t(v) \textstyle\sum_{s=t-n}^{t} \widehat{\ell}_s(i, v) .$$

Thus, we upper bound the second expectation in (5) by

$$\sum_{i \in K} \mathbb{E}\left[\eta_t(v)\,\widehat{\ell}_t(i, v) p_t(i, v) \sum_{s=t-n}^{t-1} \widehat{\ell}_s(i, v)\right] + \sum_{i \in K} \mathbb{E}\left[\eta_t(v)\,\widehat{\ell}_t(i, v)^2 p_t(i, v)\right] =: g_t^{(1)}(v) + g_t^{(2)}(v) . \qquad (7)$$

We study the two terms $g_t^{(1)}(v)$ and $g_t^{(2)}(v)$ separately. Let then $\mathcal{H}_t = \mathcal{H}_t(v)$ be the $\sigma$-algebra generated by the activations of agents and the actions drawn by them at times $1, \ldots, t-1$, and let also indicate $\mathbb{E}_t = \mathbb{E}[\cdot \mid \mathcal{H}_t]$.

First, we bound $g_t^{(1)}(v)$ in (7) using the fact that $p_t(\cdot, v)$, $\eta_t(v)$ and $b_s(\cdot, v)$ (for all $s \in \{t-n, \ldots, t\}$) are determined by the randomness in steps $1, \ldots, t-n-1$. We use the tower rule and take the conditional expectation inside since all quantities apart from $B_{t-n}(i, v), \ldots, B_t(i, v)$ are determined given $\mathcal{H}_{t-n}$, we rewrite the expression as

$$g_t^{(1)}(v) = \mathbb{E}\left[\sum_{i \in K} \eta_t(v) p_t(i, v) \sum_{s=t-n}^{t-1} \frac{\ell_t(i)}{b_t(i, v)} \frac{\ell_s(i)}{b_s(i, v)} \mathbb{E}_{t-n}\left[B_t(i, v) B_s(i, v)\right]\right] .$$

Conditional on $\mathcal{H}_{t-n}$ the Bernoulli random variables $B_s(i, v)$, and $B_t(i, v)$ for $s = t-n, \ldots, t-1$ are independent. This follows because the feedbacks at time $s$ are missing at time $t$ for $s = t-n, \ldots, t-1$, and therefore, from the independent activation of agents and the fact that the only other source of randomness is the independent internal randomization of the algorithm, they are independent random variables, implying

$$\mathbb{E}_{t-n}\left[B_t(i, v) B_s(i, v)\right] = b_t(i, v) b_s(i, v) ,$$

for $s = t-n, \ldots, t-1$. Using $\ell_t(i), \ell_s(i) \leq 1$, we then get

$$g_t^{(1)}(v) \leq \mathbb{E}\left[\sum_{i \in K} \eta_t(v) p_t(i, v) n\right] = \mathbb{E}[\eta_t(v) n] .$$

With a similar argument, we also get

$$g_t^{(2)}(v) = \mathbb{E}\left[\sum_{i \in K} \frac{\eta_t(v) \ell_t(i)^2 p_t(i, v)}{b_t(i, v)^2} \mathbb{E}_{t-n}\left[B_t(i, v)\right]\right] \leq \mathbb{E}\left[\sum_{i \in K} \eta_t(v) \frac{p_t(i, v)}{b_t(i, v)}\right] .$$

Finally, the single agent regret for each agent $v \in A$ is bounded by

$$R_T(v) \leq \mathbb{E}\left[\frac{\ln|K|}{\eta_T(v)}\right] + (n+1) + \sum_{t=n+2}^{T} \mathbb{E}\left[\eta_t(v)\left(n + \sum_{i \in K} \frac{p_t(i, v)}{b_t(i, v)}\right)\right]$$

$$= \mathbb{E}\left[\frac{\ln|K|}{\eta_T(v)}\right] + (n+1) + \sum_{t=n+2}^{T} \mathbb{E}\left[\eta_t(v) X_t(v)\right] ,$$

where, in the second line, we defined $X_t(v) = \mathbb{I}\{t > n+1\}\left(n + \sum_{i \in K} \frac{p_t(i, v)}{b_t(i, v)}\right)$.

We now take $\eta_t(v) = \sqrt{\log(K)\big/\sum_{s=1}^{t} X_s(v)}$, and we use a standard inequality stating that for any $a_t > 0$, $\sum_{t=1}^{T} a_t \big/ \sqrt{\sum_{s=1}^{t} a_s} \le 2\sqrt{\sum_{t=1}^{T} a_t}$. Applying this inequality for $a_t$ equal to $X_t(v)$ we have

$$R_T(v) \le (n+1) + \mathbb{E}\left[\sqrt{\log(K)\sum_{s=1}^{T} X_s(v)}\right] + \mathbb{E}\left[\sum_{t=1}^{T} \frac{\sqrt{\log(K)}X_t(v)}{\sqrt{\sum_{s=1}^{t} X_s(v)}}\right]$$

$$\le (n+1) + 3\mathbb{E}\left[\sqrt{\log(K)\sum_{t=1}^{T} X_t(v)}\right].$$

Multiplying by $q(v)$, summing over agents $v \in A$ we obtain

$$R_T = \sum_{v} q(v) R_T(v) = Q(n+1) + 3Q\sum_{v} \frac{q(v)}{Q}\sqrt{\sum_{t=1}^{T} \log(K)\left(n + \sum_{i \in K} \frac{p_t(i,v)}{b_t(i,v)}\right)}$$

$$\le Q(n+1) + 3Q\sqrt{\frac{\log(K)}{Q}\sum_{t=1}^{T}\left(nQ + \sum_{v}\sum_{i \in K} \frac{p_t(i,v)q(v)}{b_t(i,v)}\right)}$$

$$\le Q(n+1) + 3Q\sqrt{\log(K)\left(n + 1 + \frac{\alpha\left(N^n \boxtimes F\right)}{Q(1-e^{-1})}\right)T},$$

where the first inequality follows from Jensen's inequality and the second from Lemma 1. $\qquad\square$

Note that in Theorem 1, every agent tunes the learning rate $\eta_t(v)$ using available information at time $t$. This allows the network regret to adapt to the unknown parameters of the problems such as the time horizon $T$, the independence number $\alpha\left(N^n \boxtimes F\right)$, and the total activation mass $Q$ on $A$. This approach improves over the doubling trick approach used in Cesa-Bianchi et al. (2019) since we do not need to restart the algorithm.

## 5 Lower bound

In this section, we prove that not only the upper bound in Theorem 1 is optimal in a *minimax* sense—i.e., that it is attained for some pairs of graphs $(N, F)$—but it is also tight in an *instance-dependent* sense, for *all* pairs of graphs belonging to a large class.

**Definition 1.** *Let $\mathscr{G}$ be the class of all pairs of graphs $(N, F)$ such that $\alpha(N \boxtimes F) = \alpha(N)\alpha(F)$.*

Many sufficient conditions guaranteeing that $(N, F) \in \mathscr{G}$ are known in the graph theory literature: see, e.g., Hales (1973, Section 3), Acín et al. (2017, Theorem 6), and Rosenfeld (1967, Theorem 2). To the best of our knowledge, a full characterization of $\mathscr{G}$ is still a challenging open problem in graph theory that goes beyond the scope of this paper. It is easy to verify that if (either $N$ or) $F$ is a clique or an edgeless graph, then $(N, F) \in \mathscr{G}$. We remark that these instances cover in particular both the bandit and the full-info case that were previously only studied individually, and analyzed with *ad hoc* techniques. For some further discussion on $\mathscr{G}$, we refer the interest reader to Appendix B.1.

The proof of the lower bound in Theorem 2 exploits a reduction to a setting we introduce in Lemma 2. In this lemma, we state that in a single-agent setting with a feedback graph, if each one of $T$ time steps is independently skipped with a known and constant probability $\mu$, the learner's regret is $\Omega\big(\sqrt{\alpha(F)\mu T}\big)$. Skipped rounds do not count towards regret. More precisely, at each round $t$, there is an independent Bernoulli random variable $A_t$ with mean $\mu$. If $A_t = 0$, the learner is not required to make any predictions, incurs no loss, and receives no feedback information. The (single-agent) regret of a possibly randomized algorithm ALG on a sequence $\left(\ell_t\right)_{t \in [T]}$ of losses is defined as $R_T^{\text{sa}}(\mu, \text{ALG}, \ell) = \max_{i \in [K]} R_T^{\text{sa}}(\mu, \text{ALG}, \ell, i)$ where

$$R_T^{\text{sa}}(\mu, \text{ALG}, \ell, i) = \mathbb{E}\left[\sum_{t=1}^{T} \left(\ell_t(I_t) - \ell_t(i)\right)\mathbb{I}\{A_t = 1\}\right]$$

and $I_t$ is the random variable denoting the action played by the learner at time $t$ that only depends on the rounds $s \in \{1, \ldots, t\}$ where $A_s = 1$. The expectation in $R_T^{\text{sa}}$ is computed over both $I_t$ and $A_t$ for $t \in [T]$. Note that it is not true, in general, that $R_T^{\text{sa}}(\mu, \text{ALG}, \ell) = \mu \max_{i \in [K]} \mathbb{E}\left[\sum_{t=1}^{T} \left(\ell_t(I_t) - \ell_t(i)\right)\right]$.

**Lemma 2.** *For any feedback graph $F$, for any (possibly randomized) online learning algorithm* ALG*, for any $\mu > 0$, and for any $T \geq \max\{0.0064 \cdot \alpha(F)^3, \frac{1}{\mu^3}\}$, if each round $t \in [T]$ is independently skipped with probability $\mu$, then*

$$\inf_{\text{ALG}} \sup_{\ell} R_T^{\text{sa}}(\mu, \text{ALG}, \ell) \overset{\Omega}{=} \sqrt{\alpha(F)\mu T} \ .$$

The proof of this lemma can be found in Appendix A. Now let ALG be a possibly randomized online algorithm. Let $R_T(q, \text{ALG}, \ell)$ be the network regret (2) incurred by ALG run with oblivious network interface, losses $\ell = (\ell_t)_{t \in [T]}$, and activation probabilities $q = (q(v))_{v \in A}$. We can now prove our lower bound.

**Theorem 2.** *For any choice of $n$, any pair of graphs $(N^n, F) \in \mathscr{G}$, and all $Q \in (0, \alpha_n(N)]$, we have that for $T \geq \max\left\{0.0064 \cdot \alpha(F)^3, \alpha(N)^3/Q^3\right\}$ the following holds*

$$\inf_{\text{ALG}} \sup_{\ell, q} R_T(q, \text{ALG}, \ell) \overset{\Omega}{=} \sqrt{Q\alpha(N^n \boxtimes F)T} \ ,$$

*where the infimum is over all randomized online algorithms and the supremum is over all assignments of losses $\ell = (\ell_t)_{t \in [T]}$ and activation probabilities $(q(v))_{v \in A}$ such that $Q = \sum_{v \in A} q(v)$.*

*Proof.* Fix any $(N, n, F)$ and $Q \in (0, \alpha_n(N)]$ as in the statement of the theorem. Then $\alpha(N^n \boxtimes F) = \alpha_n(N)\alpha(F)$. Let $\mathcal{I} \subset A$ be a set of $\alpha_n(N)$ agents such that $\delta_N(u, v) > n$ for all $u, v \in \mathcal{I}$. Define the activation probabilities $q(v) = Q/\alpha_n(N) \leq 1$ for all $v \in \mathcal{I}$ and $q(v) = 0$ for all $v \in A \setminus \mathcal{I}$. By construction, no communication occurs among agents in $\mathcal{I}$. Furthermore, each agent $v$ in $\mathcal{I}$ is activated independently with the same probability $q(v) = \mathbb{P}(v \in \mathcal{A}_t) = Q/\alpha_n(N)$.

Therefore, we can use Lemma 2 to show that there exists a sequence of losses that simultaneously lower bounds the regret of all agents. Indeed, for all $T \geq \max\{0.0064 \cdot \alpha(F)^3, \alpha_n(N)^3/Q^3\}$ we have

$$
\begin{aligned}
\inf_{\text{ALG}} \sup_{\ell} R_T(q, \text{ALG}, \ell) &= \inf_{\text{ALG}} \sup_{\ell} \max_{i \in K} \sum_{v \in \mathcal{I}} \mathbb{E}\left[\sum_{t=1}^{T} \left(\ell_t(I_t(v)) - \ell_t(i)\right) \mathbb{I}\{v \in \mathcal{A}_t\}\right] \\
&= \inf_{\text{ALG}} \sup_{\ell} \max_{i \in K} \sum_{v \in \mathcal{I}} R_T^{\text{sa}}(Q/\alpha_n(N), \text{ALG}, \ell, i) \\
&= \alpha_n(N) \inf_{\text{ALG}} \sup_{\ell} R_T^{\text{sa}}(Q/\alpha_n(N), \text{ALG}, \ell) \\
&\overset{\Omega}{=} \alpha_n(N)\sqrt{\alpha(F)(Q/\alpha_n(N))T} \quad\quad\quad \text{(Lemma 2 with } \mu = Q/\alpha_n(N)\text{)} \\
&= \sqrt{Q(\alpha(F)\alpha_n(N))T} = \sqrt{Q\alpha(N^n \boxtimes F)T} \ ,
\end{aligned}
$$

where $I_t(v)$ is the random variable denoting the arm pulled at time $t$ by the instance of ALG run by agent $v$ and Lemma 2 is invoked on $|\mathcal{I}| = \alpha_n(N)$ instances of ALG with feedback graph $G = F$ and independent randomization. $\quad\square$

## 6 Experiments

To empirically appreciate the impact of cooperation, we run a number of experiments on synthetic data. Our code available at Della Vecchia (2024).

For each choice of $N$ and $F$, we compare EXP3-$\alpha^2$ run on $N$ and $F$ against a baseline which runs EXP3-$\alpha^2$ on $N'$ and $F$, where $N'$ is an edgeless communication graph. Hence, the baseline runs $A$ independent instances on the same feedback graph.

In our experiments, we fix the time horizon ($T = 10,000$), the number of arms ($K = 20$), and the number of agents ($A = 20$). We also set the delay $\delta_N$ to 1. The loss of each action is a Bernoulli random variable of parameter $1/2$, except for the optimal action which has parameter $1/2 - \sqrt{K/T}$. The activation probabilities $q(v)$ are the same for all agents $v \in A$, and range in the set $\{0.05, 0.5, 1\}$. This implies that $Q \in \{1, 10, 20\}$. The feedback graph $F$ and the communication graph $N$ are Erdős–Rényi random graphs of parameters $p_N$, $p_F \in \{0.2, 0.8\}$. For each choice of the parameters, the same realization of $N$ and $F$ was kept fixed in all the experiments, see Figure 1.

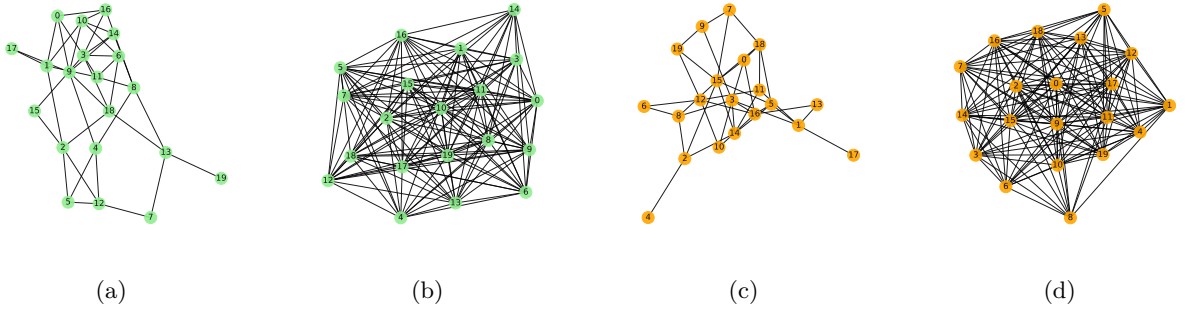

|     (a)     |     (b)     |     (c)     |     (d)     |

Figure 1: The random instances of $N$ (leftmost graphs) and $F$ (rightmost graphs) used in our experiments. The sparse graphs are Erdős–Rényi of parameter 0.2, the dense graphs are Erdős–Rényi of parameter 0.8.

In each experiment, EXP3-$\alpha^2$ and our baseline are run on the same realization of losses and agent activations. Hence, the only stochasticity left is the internal randomization of the algorithms. Our results are averages of 20 repetitions of each experiment with respect to this randomization.

| $q = 0.05$ | | | | |
|---|---|---|---|---|
| $p_N$ | $p_F$ | EXP3-$\alpha^2$ | Indep. | EXP3-$\alpha^2$/Indep. |
| 0.8 | 0.8 | 31.7 | 110.1 | 29% |
| 0.8 | 0.2 | 94.4 | 130.5 | 72% |
| 0.2 | 0.8 | 59.5 | 110.1 | 54% |
| 0.2 | 0.2 | 103 | 130.5 | 79% |

(a) Results for $q = 0.05$

| $q = 0.5$ | | | | |
|---|---|---|---|---|
| $p_N$ | $p_F$ | EXP3-$\alpha^2$ | Indep. | EXP3-$\alpha^2$/Indep. |
| 0.8 | 0.8 | 18.3 | 47.2 | 39% |
| 0.8 | 0.2 | 34.1 | 101.1 | 34% |
| 0.2 | 0.8 | 22.5 | 47.2 | 48% |
| 0.2 | 0.2 | 77.7 | 101.1 | 77% |

(b) Results for $q = 0.5$

| $q = 1$ | | | | |
|---|---|---|---|---|
| $p_N$ | $p_F$ | EXP3-$\alpha^2$ | Indep. | EXP3-$\alpha^2$/Indep. |
| 0.8 | 0.8 | 18.7 | 22.6 | 83% |
| 0.8 | 0.2 | 23.2 | 84.7 | 27% |
| 0.2 | 0.8 | 18.8 | 22.6 | 83% |
| 0.2 | 0.2 | 41.3 | 84.7 | 49% |

(c) Results for $q = 1$

Table 2: Table of the performance of EXP3-$\alpha^2$ and independent single agent optimal algorithms with feedback graphs. We summarise the performance in terms of total cumulative regret $R_T/Q$ after 1000 rounds for the two algorithms. The last column of each table is the percentage of the cumulative regret of EXP3-$\alpha^2$ with respect to the independent optimal algorithms.

Figure 2 and Table 2 summarize the results of our experiments in terms of the average regret $R_T/Q$. See Appendix D for the actual learning curves. Recall that our upper bound (1) scales with the quantity $\sqrt{\alpha(N \boxtimes F)/Q}$.

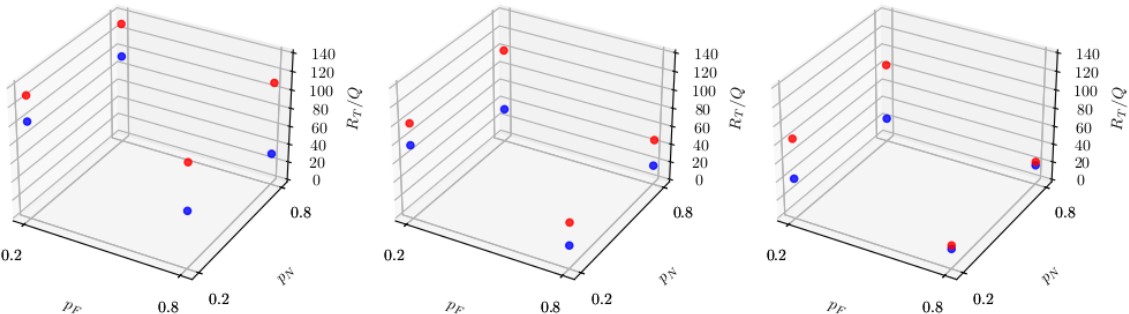

Figure 2: Average regret of EXP3-$\alpha^2$ (blue dots) against the baseline (red dots). The $X$-axis and the $Y$-axis correspond to the parameters $p_F$ and $p_N$ of the Erdős–Rényi graph, the $Z$-axis is the average regret $R_T/Q$. The three plots correspond to increasing values (from left to right) of activation probability: $q = 0.05$ (leftmost plot), $q = 0.5$ (central plot), $q = 1$ (rightmost plot).

- Note that our algorithm (blue dots) is never worse than the baseline (red dots). This is consistent with the fact that $N$ for the baseline is the edgeless graph, implying that $\alpha(N \boxtimes F) = A\,\alpha(F)$.
- Consistently with (1), the average performance gets worse when $Q \to 1$.[5]
- By construction, the performance of the baseline in each plot remains constant when $p_N$ varies in $\{0.2, 0.8\}$. On the other hand, our algorithm is worse when $N$ is sparse because $\alpha(N \boxtimes F)$ increases.
- The performance of both algorithms is worse when $F$ is sparse because, once more, $\alpha(N \boxtimes F)$ increases.

## 7 Conclusions

In this work, we nearly characterize the minimax regret in cooperative online learning with feedback graphs, showing that the dependence on $\alpha(N^n \boxtimes F)$ in our bounds is tight in all but a few, pathological instances. In a bandit setting, when all agents are active at all time steps, previous works showed that communication speeds up learning by reducing the variance of loss estimates. On the opposite end of the spectrum, in full-information settings, updating non-active agents was shown to improve regret. These results left open the question of which updates would help in intermediate settings and why. In this paper, we prove that both types of updates help local learners across the entire experts-bandits spectrum (Theorem 1). We stress that this strategy crucially depends on the stochasticity of the activations. Indeed, Cesa-Bianchi et al. (2020) disproved the naive intuition that more information automatically translates into better bounds, showing how using all the available data can lead to linear regret in the case of adversarial activations with oblivious network interface.

As we only considered undirected feedback graphs, their extension to the directed case remains open. Using the terminology introduced by Alon et al. (2015) for directed graphs, we conjecture our main result (Theorem 1) remains true in the strongly observable case. In the weakly observable case, the scaling parameter of the single-agent regret is a graph-theoretic quantity different from the independence number, and the minimax rate becomes $T^{2/3}$. In this case, we ignore the best possible scaling parameter for the network regret.

Finally, we leave open the problem of characterizing the minimax regret with stochastic activations but no oblivious network interface. We conjecture that the lower bound in Theorem 2 could be extended to the case in which the algorithms run by each agent are not necessarily instances of the same online algorithm. In other words, we conjecture that the oblivious network interface is sufficient to achieve minimax optimality.

---

[5]Also the baseline, whose agents learn in isolation, gets worse when $Q$ decreases. Indeed, when $Q = 1$ agents get only to play for $T/|A|$ time steps each, and together achieve a network regret $R_T$ that scales with $\sqrt{|A|\alpha(F)}$, as predicted by our analysis.

## Acknowledgments

NCB is partially supported by the MUR PRIN grant 2022EKNE5K (Learning in Markets and Society), funded by the NextGenerationEU program within the PNRR scheme, the FAIR (Future Artificial Intelligence Research) project, funded by the NextGenerationEU program within the PNRR-PE-AI scheme, the EU Horizon CL4-2022-HUMAN-02 research and innovation action under grant agreement 101120237, project ELIAS (European Lighthouse of AI for Sustainability). TC gratefully acknowledges the support of the University of Ottawa through grant GR002837 (Start-Up Funds) and that of the Natural Sciences and Engineering Research Council of Canada (NSERC) through grants RGPIN-2023-03688 (Discovery Grants Program) and DGECR2023-00208 (Discovery Grants Program, DGECR - Discovery Launch Supplement). RDV is thankful for the funding received by the CHIST-ERA Project Causal eXplainations in Reinforcement Learning – CausalXRL.

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

## A    Proof of Lemma 2

**Lemma 2.** *For any feedback graph $F$, for any (possibly randomized) online learning algorithm ALG, for any $\mu > 0$, and for any $T \geq \max\{0.0064 \cdot \alpha(F)^3, \frac{1}{\mu^3}\}$, if each round $t \in [T]$ is independently skipped with probability $\mu$, then*

$$\inf_{\text{ALG}} \sup_{\ell} R_T^{\text{sa}}(\mu, \text{ALG}, \ell) \stackrel{\Omega}{=} \sqrt{\alpha(F)\mu T} .$$

*Proof.* Let $\mathcal{T}$ be the (random) set of times $\{t \in [T] \mid A_t = 1\}$ and let $\tau_1 < \tau_2 < \cdots < \tau_{|\mathcal{T}|}$ the (random) elements of $\mathcal{T}$ in increasing order. Fix a (possibly randomized) online learning algorithm ALG and a sequence $\ell = (\ell_t)_{t \in [T]}$ of losses. For any random variable $J$ (later, $J$ and the corresponding "hard" instance will be those used in the lower bound for online learning with feedback graphs: (Alon et al., 2017, Theorem 5)). Let $N_t = A_1 + \cdots + A_{t-1}$ be the number of rounds actually played up to time $t$. Then

$$R_T(\mu, \text{ALG}, \ell) = \max_{i \in [K]} \mathbb{E}\left[\sum_{t=1}^{T} \big(\ell_t(I_{N_t+1}) - \ell_t(i)\big)\mathbb{I}\{A_t = 1\}\right] = \max_{i \in [K]} \mathbb{E}\left[\sum_{s \in [|\mathcal{T}|]} \big(\ell_{\tau_s}(I_s) - \ell_{\tau_s}(i)\big)\right]$$

$$\geq \mathbb{E}\left[\sum_{s \in [|\mathcal{T}|]} \big(\ell_{\tau_s}(I_s) - \ell_{\tau_s}(J)\big)\right]$$

$$= \sum_{n \in [T]} \sum_{\substack{\mathcal{T}_0 \subset [T] \\ |\mathcal{T}_0| = n}} \mathbb{E}\left[\sum_{s \in [|\mathcal{T}|]} \big(\ell_{\tau_s}(I_s) - \ell_{\tau_s}(J)\big) \mid \mathcal{T} = \mathcal{T}_0, |\mathcal{T}_0| = n\right] \mathbb{P}\left(\mathcal{T} = \mathcal{T}_0, |\mathcal{T}_0| = n\right) .$$

Then, we recognize that the conditional expectation in the previous formula is the expected regret for single-agent online learning with feedback graph. Therefore, from (Alon et al., 2017, Theorem 5) we get that, letting $C_1 = (8/100)^2$ and , for all $T \geq C_1\alpha(F)^3$,

$$\inf_{\text{ALG}} \sup_{\ell} R_T(\mu, \text{ALG}, \ell) \geq \sum_{n \in [T]} \sum_{\substack{\mathcal{T}_0 \subset [T] \\ |\mathcal{T}_0| = n}} \left(\varepsilon n\left(\frac{1}{2} - 2\varepsilon\sqrt{\frac{n}{\alpha(F)}}\right)\right) \mathbb{P}\left(\mathcal{T} = \mathcal{T}_0, |\mathcal{T}_0| = n\right)$$

$$= \sum_{n \in [T]} \left(\varepsilon n\left(\frac{1}{2} - 2\varepsilon\sqrt{\frac{n}{\alpha(F)}}\right)\right) \mathbb{P}\left(|\mathcal{T}| = n\right)$$

$$= \sum_{n \in [T]} \varepsilon n\left(\frac{1}{2} - 2\varepsilon\sqrt{\frac{n}{\alpha(F)}}\right) f_{\text{Bin}(\mu, T)}(n) ,$$

where $f_{\text{Bin}(\mu, T)}$ is the p.m.f. of a Binomial random variable with parameters $p, T$. We want $\varepsilon = \varepsilon^*(p, T)$ that maximizes that expression. Therefore, by defining $g(\varepsilon) = a\varepsilon + b\varepsilon^2$ as the following quadratic polynomial in $\varepsilon$

$$g(\varepsilon) = \sum_{n \in [T]} \left(\varepsilon n\left(\frac{1}{2} - 2\varepsilon\sqrt{\frac{n}{\alpha(F)}}\right)\right) f_{\text{Bin}(\mu, T)}(n)$$

$$= \sum_{n \in [T]} \left(\frac{n}{2} f_{\text{Bin}(\mu, T)}(n)\right) \varepsilon - 2 \sum_{m \in [T]} \left(\frac{m^{3/2}}{\alpha(F)^{1/2}} f_{\text{Bin}(\mu, T)}(m)\right) \varepsilon^2 ,$$

where $a = \sum_{n \in [T]} \left(\frac{n}{2} f_{\text{Bin}(\mu, T)}(n)\right)$ and $b = -2\sum_{m \in [T]} \left(\frac{m^{3/2}}{\alpha(F)^{1/2}} f_{\text{Bin}(\mu, T)}(m)\right)$. We find that the maximum of $g$ is achieved in $-\frac{a}{2b}$, which gives the optimal value

$$\varepsilon^*(p, T) = \frac{\sum_{n \in [T]} \left(\frac{n}{2} f_{\text{Bin}(\mu, T)}(n)\right)}{4 \sum_{m \in [T]} \left(\frac{m^{3/2}}{\alpha(F)^{1/2}} f_{\text{Bin}(\mu, T)}(m)\right)}$$

and the function $g$ evaluated at the optimal value is equal to $g^* = -\frac{a^2}{4b}$, i.e.,

$$g^* = g(\varepsilon^*(p,T)) = g\left(\frac{\sum_{n\in[T]}\left(\frac{n}{2}f_{\mathrm{Bin}(\mu,T)}(n)\right)}{4\sum_{m\in[T]}\left(\frac{m^{3/2}}{\alpha(F)^{1/2}}f_{\mathrm{Bin}(\mu,T)}(m)\right)}\right) = \frac{\sqrt{\alpha(F)}}{8}\frac{\left(\sum_{n\in[T]}nf_{\mathrm{Bin}(\mu,T)}(n)\right)^2}{\sum_{m\in[T]}\left(m^{3/2}f_{\mathrm{Bin}(\mu,T)}(m)\right)}\,.$$

Therefore, we can lower bound the regret with $g^*$ and obtain

$$\inf_{\mathrm{ALG}}\sup_{\ell}R_T(\mu,\mathrm{ALG},\ell) \geq \frac{\sqrt{\alpha(F)}}{8}\frac{\left(\sum_{n\in[T]}nf_{\mathrm{Bin}(\mu,T)}(n)\right)^2}{\sum_{m\in[T]}\left(m^{3/2}f_{\mathrm{Bin}(\mu,T)}(m)\right)} = \frac{\sqrt{\alpha(F)\mu T}}{8}\frac{(\mu T)^2}{\sum_{m\in[T]}\left(m^{3/2}f_{\mathrm{Bin}(\mu,T)}(m)\right)}$$

$$= \frac{\sqrt{\alpha(F)\mu T}}{8}\frac{(\mu T)^{3/2}}{\sum_{m\in[T]}\left(m^{3/2}f_{\mathrm{Bin}(\mu,T)}(m)\right)}\,.$$

where in the first equality we substituted the expected value of a binomial distribution of parameter $\mu$.

We now want to prove the existence of a constant $c > 0$ such that, for every $\mu > 0$ and every $T \geq \frac{1}{\mu^3}$

$$\frac{(\mu T)^{3/2}}{\sum_{m\in[T]}\left(m^{3/2}f_{\mathrm{Bin}(\mu,T)}(m)\right)} \geq c\,, \quad\text{or equivalently}\quad \frac{\sum_{m\in[T]}\left(m^{3/2}f_{\mathrm{Bin}(\mu,T)}(m)\right)}{(\mu T)^{3/2}} \leq c\,.$$

We split the sum over $m \in [T]$ into two blocks, the first for $1 \leq m \leq c_2\lfloor\mu T\rfloor$ and the second for $c_2\lfloor\mu T\rfloor < m \leq T$ for a constant $c_2 = \left\lceil\frac{\mu T}{\lfloor\mu T\rfloor}\left(\frac{1}{\mu}\left(\sqrt{\frac{1}{2T}\ln\left(\frac{T^{3/2}}{c_1}\right)}-\frac{1}{T}\right)+1\right)\right\rceil$ and with $c_1 > 0$:

$$\frac{\sum_{m\in[T]}\left(m^{3/2}f_{\mathrm{Bin}(\mu,T)}(m)\right)}{(\mu T)^{3/2}} = \frac{\sum_{m=1}^{c_2\lfloor\mu T\rfloor}\left(m^{3/2}f_{\mathrm{Bin}(\mu,T)}(m)\right)}{(\mu T)^{3/2}} + \frac{\sum_{m>c_2\lfloor\mu T\rfloor}\left(m^{3/2}f_{\mathrm{Bin}(\mu,T)}(m)\right)}{(\mu T)^{3/2}}\,. \quad (8)$$

The idea is to choose the split point $c_2\lfloor\mu T\rfloor$ so that we can upper bound the tail mass using Hoeffding's inequality. Hoeffding's inequality yields the simple bound $F_{\mathrm{Bin}(\mu,T)}(m) \leq \exp\left(-2T\left(\mu-\frac{m}{T}\right)^2\right)$, and together with symmetry properties of the binomial distribution $1 - F_{\mathrm{Bin}(\mu,T)}(m) = F_{\mathrm{Bin}(1-\mu,T)}(T-m)$ we obtain a bound on the upper tail. This contribution compensates exactly the term $\mu^{3/2}$ left at the denominator for $T \geq 1/\mu$, leaving just the constant $c_1$:

$$\frac{\sum_{m>c_2\lfloor\mu T\rfloor}\left(m^{3/2}f_{\mathrm{Bin}(\mu,T)}(m)\right)}{(\mu T)^{3/2}} \leq \frac{e^{-2T\left((1-\mu)-\frac{T-(c_2\lfloor\mu T\rfloor+1)}{T}\right)^2}T^{3/2}}{(\mu T)^{3/2}}$$

$$= \frac{e^{-2T\left(\mu\left(c_2\frac{\lfloor\mu T\rfloor}{\mu T}-1\right)+\frac{1}{T}\right)^2}}{\mu^{3/2}}$$

$$= \frac{c_1}{(\mu T)^{3/2}}$$

$$\leq c_1\,.$$

For the first term in Equation (8), we upper bound the lower tail simply by one:

$$\sum_{m=1}^{c_2\lfloor\mu T\rfloor}\left(m^{3/2}f_{\mathrm{Bin}(\mu,T)}(m)\right) \leq c_2\lfloor\mu T\rfloor \cdot F_{\mathrm{Bin}(\mu,T)}(m) \leq c_2\lfloor\mu T\rfloor\,.$$

We conclude by proving that the first term in Equation (8) is bounded by a constant. If we take $m \leq c_2\lfloor\mu T\rfloor$ we obtain for $T \geq 2/\mu$ and $c_1 \geq 1$

$$\frac{\sum_{m=1}^{c_2\lfloor\mu T\rfloor}\left(m^{3/2}f_{\mathrm{Bin}(\mu,T)}(m)\right)}{(\mu T)^{3/2}} \leq \frac{(c_2\lfloor\mu T\rfloor)^{3/2}}{(\mu T)^{3/2}}$$

$$\leq (c_2)^{3/2}$$

$$\leq \left(1 + \frac{\mu T}{\lfloor \mu T \rfloor} \left(\frac{1}{\mu}\left(\sqrt{\frac{1}{2T}\ln\left(\frac{T^{3/2}}{c_1}\right)} - \frac{1}{T}\right) + 1\right)\right)^{3/2}$$

$$\leq \left(2 + \frac{3\sqrt{3}}{8}\frac{1}{\mu\sqrt{T}}\left(\sqrt{\log(T)}\right)\right)^{3/2}$$

$$\leq \left(2 + \frac{3\sqrt{3}}{8}\right)^{3/2}\left(\frac{1}{\mu\sqrt{T}}\sqrt{\ln\left(\frac{T^{3/2}}{c_1}\right)}\right)^{3/2}$$

$$\leq 4.32\left(\frac{1}{\mu\sqrt{T}}\sqrt{\ln\left(\frac{T^{3/2}}{c_1}\right)}\right)^{3/2}$$

$$\leq 4.32\left(\ln T\right)^{3/4}\frac{1}{(\mu^2 T)^{3/4}}$$

$$\leq 4.32\frac{(\ln T)^{3/4}}{T^{1/4}}$$

$$\leq 4.32 \cdot 1.08 \leq 5\,,$$

where in the third-last inequality we used $\mu \geq \frac{1}{T^{1/3}}$. Putting everything together and letting $c_1 = 1$ yields

$$\inf_{\text{ALG}}\sup_{\ell} R_T(\mu, \text{ALG}, \ell) \geq \frac{3}{4}\sqrt{\alpha(F)\mu T}\,.$$

$\square$

## B  Graph-theoretic results

In this section, we present a general version of a graph-theoretic lemma (Lemma 6) that is crucial for our positive results in Section 4. Before stating it, we recall a few known results.

The first result is a direct consequence of (Alon et al., 2017, Lemma 10) specialized to undirected graphs.

**Lemma 3.** *Let $\mathcal{G} = (\mathcal{V}, \mathcal{E})$ be an undirected graph containing all self-loops and $\alpha_d(\mathcal{G})$ its d-th independence number. For all $i \in \mathcal{V}$, let $\mathcal{N}_d^{\mathcal{G}}(i)$ be the d-th neighborhood of $i$, $p(i) \geq 0$, and $P(i) = \sum_{j \in \mathcal{N}_d^{\mathcal{G}}(i)} p(j) > 0$. Then*

$$\sum_{i \in \mathcal{V}}\frac{p(i)}{P(i)} \leq \alpha_d(\mathcal{G})\,.$$

*Proof.* Initialize $V_1 = \mathcal{V}$, fix $j_1 \in \arg\min_{j \in V_1} P(j)$, and denote $V_2 = \mathcal{V} \setminus \mathcal{N}(j_1)$. For $k \geq 2$ fix $j_k \in \arg\min_{j \in V_k} P(j)$ and shrink $V_{k+1} = V_k \setminus \mathcal{N}(j_k)$ until $V_{k+1} = \varnothing$. Since $\mathcal{G}$ is undirected $j_k \notin \bigcup_{s=1}^{k-1}\mathcal{N}(j_s)$, therefore the number $m$ of times that an action can be picked this way is upper bounded by $\alpha$. Denoting $\mathcal{N}'(j_k) = V_k \cap \mathcal{N}(j_k)$ this implies

$$\sum_{i \in \mathcal{V}}\frac{p(i)}{P(i)} = \sum_{k=1}^{m}\sum_{i \in \mathcal{N}'(j_k)}\frac{p(i)}{P(i)} \leq \sum_{k=1}^{m}\sum_{i \in \mathcal{N}'(j_k)}\frac{p(i)}{P(j_k)} \leq \sum_{k=1}^{m}\frac{\sum_{i \in \mathcal{N}(j_k)}p(i)}{P(j_k)} = m \leq \alpha\,.$$

concluding the proof. $\square$

The following result, known as the inequality of arithmetic and geometric means, or simply AM-GM inequality, is used in the proofs of Lemmas 1, 5, and 6.

**Lemma 4** (AM-GM inequality). *For any $x_1, \ldots, x_r \in [0, \infty)$,*

$$\frac{x_1 + \cdots + x_r}{r} \geq (x_1 \cdot \cdots \cdot x_r)^{1/r}\,.$$

*Proof.* By Jensen's inequality,

$$\ln\left(\frac{1}{r}\sum_{i=1}^{r}x_i\right) \geq \sum_{i=1}^{r}\frac{1}{r}\ln(x_i) = \sum_{i=1}^{r}\ln\left(x_i^{1/r}\right) = \ln\left(\prod_{i=1}^{r}x_i^{1/r}\right) .$$

$\square$

The second result is proven in (Cesa-Bianchi et al., 2019, Lemma 3), but here we give a slightly different proof based on the AM-GM inequality.

**Lemma 5.** *Let $\mathcal{G} = (\mathcal{V}, \mathcal{E})$ be an undirected graph containing all self-loops and $\alpha_d(\mathcal{G})$ its d-th independence number. For all $v \in \mathcal{V}$, let $\mathcal{N}_d^{\mathcal{G}}(v)$ be the d-th neighborhood of $v$, $c(v) \geq 0$, and $C(v) = 1 - \prod_{w \in \mathcal{N}_d^{\mathcal{G}}(v)}(1 - c(w)) > 0$. Then*

$$\sum_{v \in \mathcal{V}}\frac{c(v)}{C(v)} \leq \frac{1}{1 - e^{-1}}\left(\alpha_d(\mathcal{G}) + \sum_{v \in \mathcal{V}}c(v)\right) .$$

*Proof.* Set for brevity $P(v) = \sum_{w \in \mathcal{N}_d^{\mathcal{G}}(v)}c(w)$. Then we can write

$$\sum_{v \in \mathcal{V}}\frac{c(v)}{C(v)} = \underbrace{\sum_{v \in \mathcal{V}\,:\,P(v)\geq 1}\frac{c(v)}{C(v)}}_{(I)} + \underbrace{\sum_{v \in \mathcal{V}\,:\,P(v)<1}\frac{c(v)}{C(v)}}_{(II)} ,$$

and proceed by upper bounding the two terms (I) and (II) separately. Let $r(v)$ be the cardinality of $\mathcal{N}_d^{\mathcal{G}}(v)$. We have, for any given $v \in \mathcal{V}$,

$$\min\left\{C(v) : \sum_{w \in \mathcal{N}(v)}c(w) \geq 1\right\} = \min\left\{C(v) : \sum_{w \in \mathcal{N}(v)}c(w) = 1\right\}$$

$$= 1 - \max\left\{\prod_{w \in \mathcal{N}_d^{\mathcal{G}}(v)}(1 - c(w)) : \sum_{w \in \mathcal{N}(v)}(1 - c(w)) = r(v) - 1\right\}$$

$$\geq 1 - \left(1 - \frac{1}{r(v)}\right)^{r(v)} \geq 1 - e^{-1} .$$

where the first equality follows from the definition of $C(v)$ and the monotonicity of $x \mapsto 1 - x$, the first inequality is implied by the AM-GM inequality (Lemma 4), and the last one comes from $r(v) \geq 1$ (for $v \in \mathcal{N}_d^{\mathcal{G}}(v)$). Hence

$$(I) \leq \sum_{v \in \mathcal{V}\,:\,P(v)\geq 1}\frac{c(v)}{1 - e^{-1}} \leq \sum_{v \in \mathcal{V}}\frac{c(v)}{1 - e^{-1}} .$$

As for (II), using the inequality $1 - x \leq e^{-x}$, $x \in \mathbb{R}$, with $x = c(w)$, we can write

$$C(v) \geq 1 - \exp\left(-\sum_{w \in \mathcal{N}_d^{\mathcal{G}}(v)}c(w)\right) = 1 - \exp(-P(v)) .$$

Now, since in (II) we are only summing over $v$ such that $P(v) < 1$, we can use the inequality $1 - e^{-x} \geq (1 - e^{-1})x$, holding when $x \in [0,1]$, with $x = P(v)$, thereby concluding that

$$C(v) \geq (1 - e^{-1})P(v) .$$

Thus

$$(II) \leq \sum_{v \in \mathcal{V}\,:\,P(v)<1}\frac{c(v)}{(1 - e^{-1})P(v)} \leq \frac{1}{1 - e^{-1}}\sum_{v \in \mathcal{V}}\frac{c(v)}{P(v)} \leq \frac{\alpha}{1 - e^{-1}} ,$$

where in the last step we used Lemma 3. $\square$

We can now state a more general version of our key graph-theoretic result, which can be proved similarly to Lemma 1.

**Lemma 6.** *Let $\mathcal{G}_1 = (\mathcal{V}_1, \mathcal{E}_1)$ and $\mathcal{G}_2 = (\mathcal{V}_2, \mathcal{E}_2)$ be two undirected graphs containing all self-loops and $\alpha\big(\mathcal{G}_1 \boxtimes \mathcal{G}_2\big)$ the independence number of their strong product $\mathcal{G}_1 \boxtimes \mathcal{G}_2$. For all $(i,j) \in \mathcal{V}_1 \times \mathcal{V}_2$, let also $\mathcal{N}_1^{\mathcal{G}_1}(i)$, $\mathcal{N}_1^{\mathcal{G}_2}(v)$, and $\mathcal{N}_1^{\mathcal{G}_1 \boxtimes \mathcal{G}_2}(i,v)$ be the first neighborhoods of $i$ (in $\mathcal{G}_1$), $v$ (in $\mathcal{G}_2$), and $(i,v)$ (in $\mathcal{G}_1 \boxtimes \mathcal{G}_2$). If $\boldsymbol{w} = \big(w(j,u)\big)_{(j,u) \in \mathcal{V}_1 \times \mathcal{V}_2}$ is an arbitrary matrix with non-negative entries such that $1 - \sum_{j \in \mathcal{N}_1^{\mathcal{G}_1}(i)} w(j,u) \geq 0$ for all $(i,u) \in \mathcal{V}_1 \times \mathcal{V}_2$ and $1 - \prod_{u \in \mathcal{N}_1^{\mathcal{G}_2}(v)} \big(1 - \sum_{j \in \mathcal{N}_1^{\mathcal{G}_1}(i)} w(j,u)\big) > 0$ for all $(i,v) \in \mathcal{V}_1 \times \mathcal{V}_2$, then*

$$\sum_{i \in \mathcal{V}_1} \sum_{v \in \mathcal{V}_2} \frac{w(i,v)}{1 - \prod_{u \in \mathcal{N}_1^{\mathcal{G}_2}(v)} \big(1 - \sum_{j \in \mathcal{N}_1^{\mathcal{G}_1}(i)} w(j,u)\big)} \leq \frac{e}{e-1} \left( \alpha\big(\mathcal{G}_1 \boxtimes \mathcal{G}_2\big) + \sum_{i \in \mathcal{V}_1} \sum_{v \in \mathcal{V}_2} w(i,v) \right) \, .$$

### B.1 Further discussion on $\mathscr{G}$

In general, $\alpha(N)\alpha(F) \leq \alpha\big(N \boxtimes F\big)$ holds for any arbitrary pairs of graphs $N, F$. Indeed, the Cartesian product $I \times J$ of an independent set $I$ of $N$ and an independent set $J$ of $F$ is an independent set of $N \boxtimes F$. There exist graphs $N, F$ with $\alpha(N)\alpha(F) \ll \alpha\big(N \boxtimes F\big)$, but these appear to be quite rare and pathological cases. For the sake of completeness, we add an example of such a construction below. This shows that not all pairs of graphs belong to $\mathscr{G}$.

**Example 1.** *Take as the first graph $G_1 = (V_1, E_1)$, the cycle $C_5$ over $5$ vertices. Then, for any $k \geq 2$, build $G_k = (V_k, E_k)$ inductively by replacing each vertex $v \in V_{k-1}$ by a copy of $C_5$ and each edge $e \in E_{k-1}$ by a copy of $K_{5,5}$ (the complete bipartite graph with partitions of size $5$ and $5$) between the two copies of $C_5$ that replaced its endpoints. It can be shown that $\alpha(G_k) = 2^k$ but $\alpha\big(G_k \boxtimes G_k\big) \geq 5^k \gg 4^k = \alpha(G_k)^2$.*

*To see why, note first that $\alpha(C_5) = 2$ but $\alpha\big(C_5 \boxtimes C_5\big) \geq 5$, by choosing the independent set containing the $5$ vertices $(1,1), (2,3), (3,5), (4,2), (5,4)$. For $k \geq 2$, $\alpha(G_k) = 2^k$ but we can take the analogous in $G_k \boxtimes G_k$ of the above independent set in $C_5 \boxtimes C_5$. This gives $5$ sets $S_1, S_2, S_3, S_4, S_5$ of $25^{k-1}$ vertices each, with no edges between $S_i, S_j$ when $i \neq j$. The subgraph of $G_k \boxtimes G_k$ induced by each $S_i$ is simply the previous iteration $G_{k-1}$ of this construction, and proceeding by induction we can find an independent subset of each $S_i$ with $5^{k-1}$ vertices, giving a total of $5^k$ independent vertices.*

## C The upper bound of Cesa-Bianchi et al. (2020) for experts

(Cesa-Bianchi et al., 2020, Theorem 10) gives theoretical guarantees for the *average* regret over active agents. In this section, we briefly discuss how to convert their statement to a corresponding result for the *total* regret over active agents that is the focus of our present work.

Before stating the theorem, we recall that the *convex conjugate* $f^* \colon \mathbb{R}^d \to \mathbb{R}$ of a convex function $f \colon \mathbb{X} \to \mathbb{R}$ is defined, for any $\boldsymbol{x} \in \mathbb{R}^d$, by $f^*(\boldsymbol{x}) = \sup_{\boldsymbol{w} \in \mathbb{X}} \big(\boldsymbol{x} \cdot \boldsymbol{w} - f(\boldsymbol{w})\big)$. Moreover, given $\sigma > 0$, we say that $f$ is $\sigma$-*strongly convex* on $\mathbb{X}$ with respect to a norm $\|\cdot\|$ if, for all $\boldsymbol{u}, \boldsymbol{w} \in \mathbb{X}$, we have $f(\boldsymbol{u}) \geq f(\boldsymbol{w}) + \nabla f(\boldsymbol{w}) \cdot (\boldsymbol{u} - \boldsymbol{w}) + \frac{\sigma}{2} \|\boldsymbol{u} - \boldsymbol{w}\|^2$. The following well-known result can be found in (Shalev-Shwartz et al., 2012, Lemma 2.19 and subsequent paragraph).

**Lemma 7.** *Let $f \colon \mathbb{X} \to \mathbb{R}$ be a strongly convex function on $\mathbb{X}$. Then the convex conjugate $f^*$ is everywhere differentiable on $\mathbb{R}^d$.*

The following result—see, e.g., (Orabona et al., 2015, bound (6) in Corollary 1 with $F$ set to zero)—shows an upper bound on the regret of Algorithm 2 for single-agent online convex optimization with expert feedback.

**Theorem 3.** *Let $g \colon \mathbb{X} \to \mathbb{R}$ be a differentiable function $\sigma$-strongly convex with respect to $\|\cdot\|$. Then the regret of Algorithm 2 run with $g_t = \frac{\sqrt{t}}{\eta} g$, for $\eta > 0$, satisfies*

$$\sum_{t=1}^{T} \ell_t\big(\boldsymbol{x}_t\big) - \inf_{\boldsymbol{x} \in \mathbb{X}} \sum_{t=1}^{T} \ell_t\big(\boldsymbol{x}\big) \leq \frac{D}{\eta} \sqrt{T} + \frac{\eta}{2\sigma} \sum_{t=1}^{T} \frac{1}{\sqrt{t}} \left\| \nabla \ell_t \right\|_*^2 \, ,$$

---

**Algorithm 2:**

---

**input:** $\sigma_t$-strongly convex regularizers $g_t \colon \mathbb{X} \to \mathbb{R}$ for $t = 1, 2, \dots$
**initialization:** $\boldsymbol{\theta}_1 = \mathbf{0} \in \mathbb{R}^d$
**for** $t = 1, 2, \dots$ **do**
    choose $\boldsymbol{w}_t = \nabla g_t^*(\boldsymbol{\theta}_t)$
    observe $\nabla \ell_t(\boldsymbol{w}_t) \in \mathbb{R}^d$
    update $\boldsymbol{\theta}_{t+1} = \boldsymbol{\theta}_t - \nabla \ell_t(\boldsymbol{w}_t)$

---

*where $D = \sup g$ and $\|\cdot\|_*$ is the dual norm of $\|\cdot\|$. If $\sup \|\nabla \ell_t\|_* \leq L$, then choosing $\eta = \sqrt{2\sigma D}/L$ gives $R_T \leq L\sqrt{2DT/\sigma}$.*

We can now present the equivalent of (Cesa-Bianchi et al., 2020, Theorem 10) for cooperative online covenx optimization with expert feedback (i.e., $F$ is a clique) where $n = 1$ but the feedback is broadcast to first neighbor immediately after an action is played (rather than the following round).

**Theorem 4.** *Consider a network $N = (A, E_N)$ of agents. If all agents $v$ run Algorithm 2 with an oblivious network interface and $g_t = \frac{\sqrt{t}}{\eta} g$, where $\|g_t\|_*$ is upper bounded by a constant $L > 0$, $\eta > 0$ is a learning rate, and the regularizer $g \colon \mathbb{X} \to \mathbb{R}$ is differentiable, $\sigma$-strongly convex with respect to some norm $\|\cdot\|$, and upper bounded by a constant $M^2$, then the network regret satisfies*

$$R_T \leq \left( \frac{M^2}{\eta} + \frac{\eta L^2}{2\sigma} \right) \sqrt{2Q(\alpha(N) + Q)T} \ .$$

*For $\eta = \sqrt{2\sigma} M/L$, we have*

$$R_T \leq \left( \sqrt{2\sigma} LM \right) \sqrt{Q(\alpha(N) + Q)T} \ .$$

*Proof sketch.* For any $\boldsymbol{x} \in \mathbb{X}$, agent $v$, and time $t$, let $\boldsymbol{x}_t(v)$ be the prediction made by $v$ at time $t$, $r_t(v, \boldsymbol{x}) = \ell_t(\boldsymbol{x}_t(v)) - \ell_t(\boldsymbol{x})$, $Q_v = \Pr\left( v \in \bigcup_{w \in \mathcal{A}_t} \mathcal{N}_1^N(w) \right) = 1 - \prod_{w \in \mathcal{N}_1^N(v)} (1 - q(w))$, and $A' := \{ w \in A : q(w) > 0 \}$. Proceeding as in (Cesa-Bianchi et al., 2020, Theorem 2) yields, for each $v \in A'$ and $\boldsymbol{x} \in \mathbb{X}$,

$$\mathbb{E}\left[ \sum_{t=1}^{T} r_t(v, \boldsymbol{x}) \right] \leq \left( \frac{M^2}{\eta} + \frac{\eta L^2}{2} \right) \sqrt{\frac{T}{Q_v}} \ . \tag{9}$$

Now, by the independence of the activations of the agents at time $t$ and $\left( r_t(v, \boldsymbol{x}) \right)_{v \in A', \boldsymbol{x} \in \mathbb{X}}$, we get

$$R_T = \sup_{\boldsymbol{x} \in \mathbb{X}} \sum_{v \in V'} q(v) \sum_{t=1}^{T} \mathbb{E}\left[ r_t(v, \boldsymbol{x}) \right] \ . \tag{10}$$

Putting Equations (9) and (10) together and applying Jensen's inequality yields

$$R_T \leq \left( \sum_{v \in V'} q(v) \sqrt{\frac{1}{Q_v}} \right) \left( \frac{M^2}{\eta} + \frac{\eta L^2}{2} \right) \sqrt{T} \leq \sqrt{Q \sum_{v \in V'} \frac{q_v}{Q_v}} \left( \frac{M^2}{\eta} + \frac{\eta L^2}{2} \right) \sqrt{T} \ .$$

The proof is concluded by invoking Lemma 5. $\qquad \square$

## D   Learning curves

We also plot the average regret $R_T/Q$ against the number $T$ of rounds. Our algorithm is the blue curve and the baseline is the red curve. Recall that these curves are averages over 20 repetitions of the same experiment (the shaded areas correspond to one standard deviation) where the stochasticity is due to the internal randomization of the algorithms. Experiments are designed to show the difference in performance when we allow agents to communicate and when we do not. The strong product captures in a mathematical

form this difference in the regret bound for our algorithm, while the experiments here show it empirically. In particular, the bound for the case of no communication is bigger, and performances are worse in our simulations, as expected from theory.

Experiments were run on a local cluster of CPUs (Intel Xeon E5-2623 v3, 3.00GHz), parallelizing the code over four cores. The run took approximately two hours.

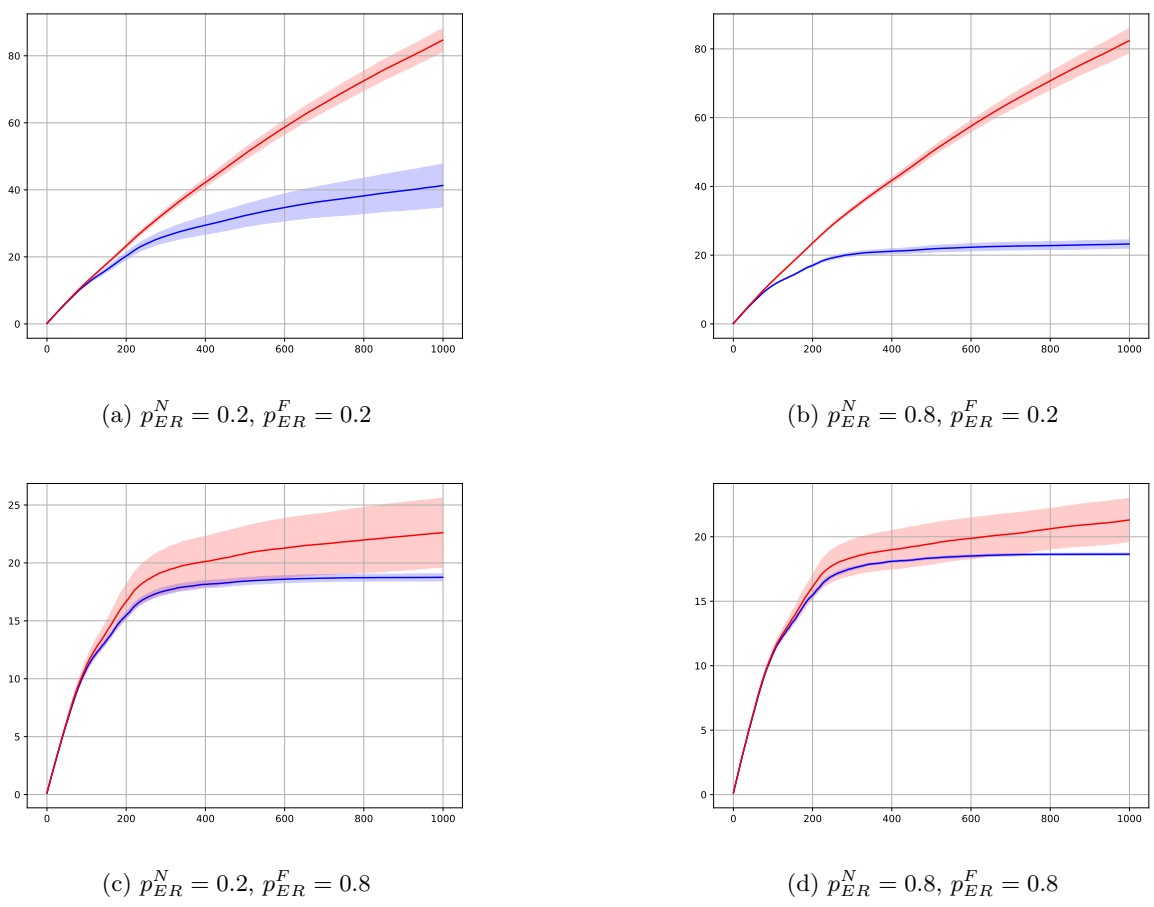

(a) $p_{ER}^N = 0.2$, $p_{ER}^F = 0.2$

(b) $p_{ER}^N = 0.8$, $p_{ER}^F = 0.2$

(c) $p_{ER}^N = 0.2$, $p_{ER}^F = 0.8$

(d) $p_{ER}^N = 0.8$, $p_{ER}^F = 0.8$

Figure 3: Average regret $R_T/Q$ against $T = 1000$ of rounds. Activation probability $q = 1$.

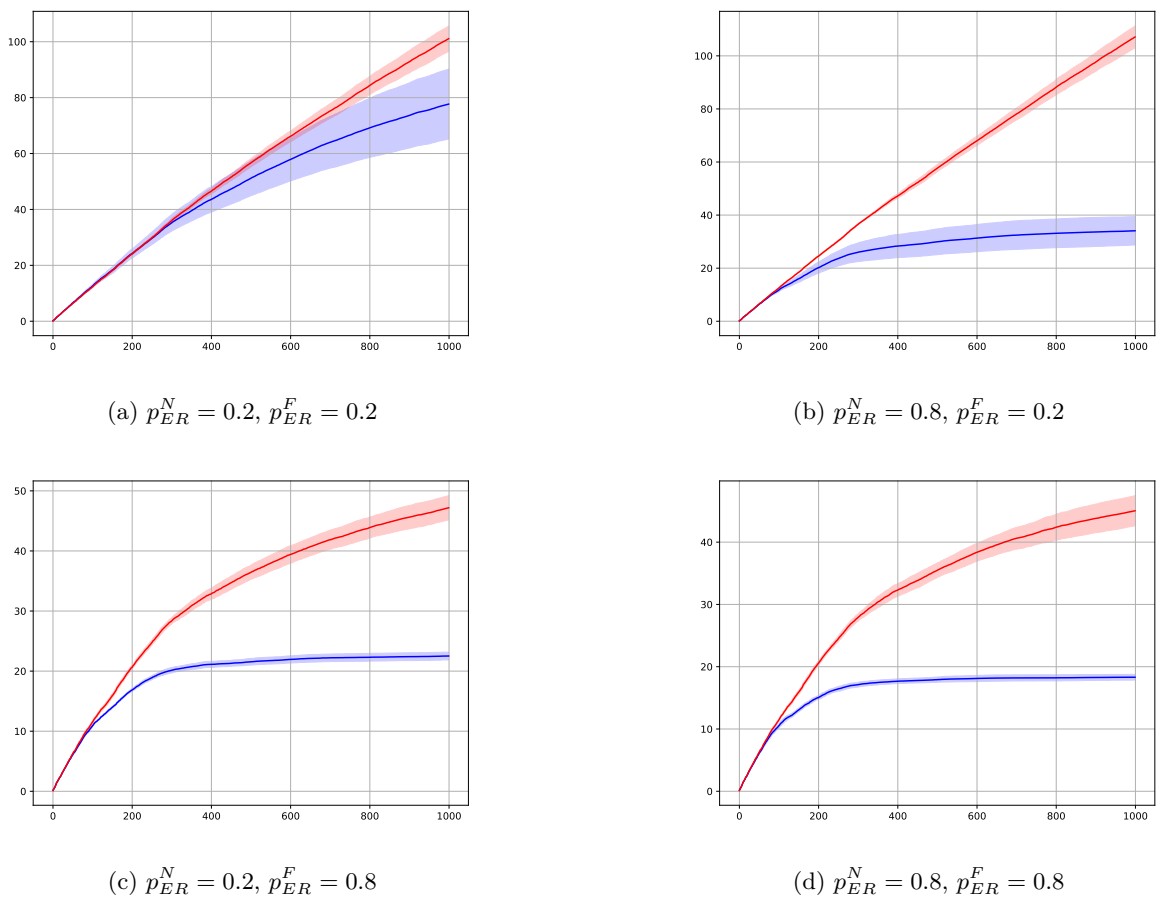

(a) $p_{ER}^N = 0.2$, $p_{ER}^F = 0.2$

(b) $p_{ER}^N = 0.8$, $p_{ER}^F = 0.2$

(c) $p_{ER}^N = 0.2$, $p_{ER}^F = 0.8$

(d) $p_{ER}^N = 0.8$, $p_{ER}^F = 0.8$

Figure 4: Average regret $R_T/Q$ against $T = 1000$ of rounds. Activation probability $q = 0.5$.

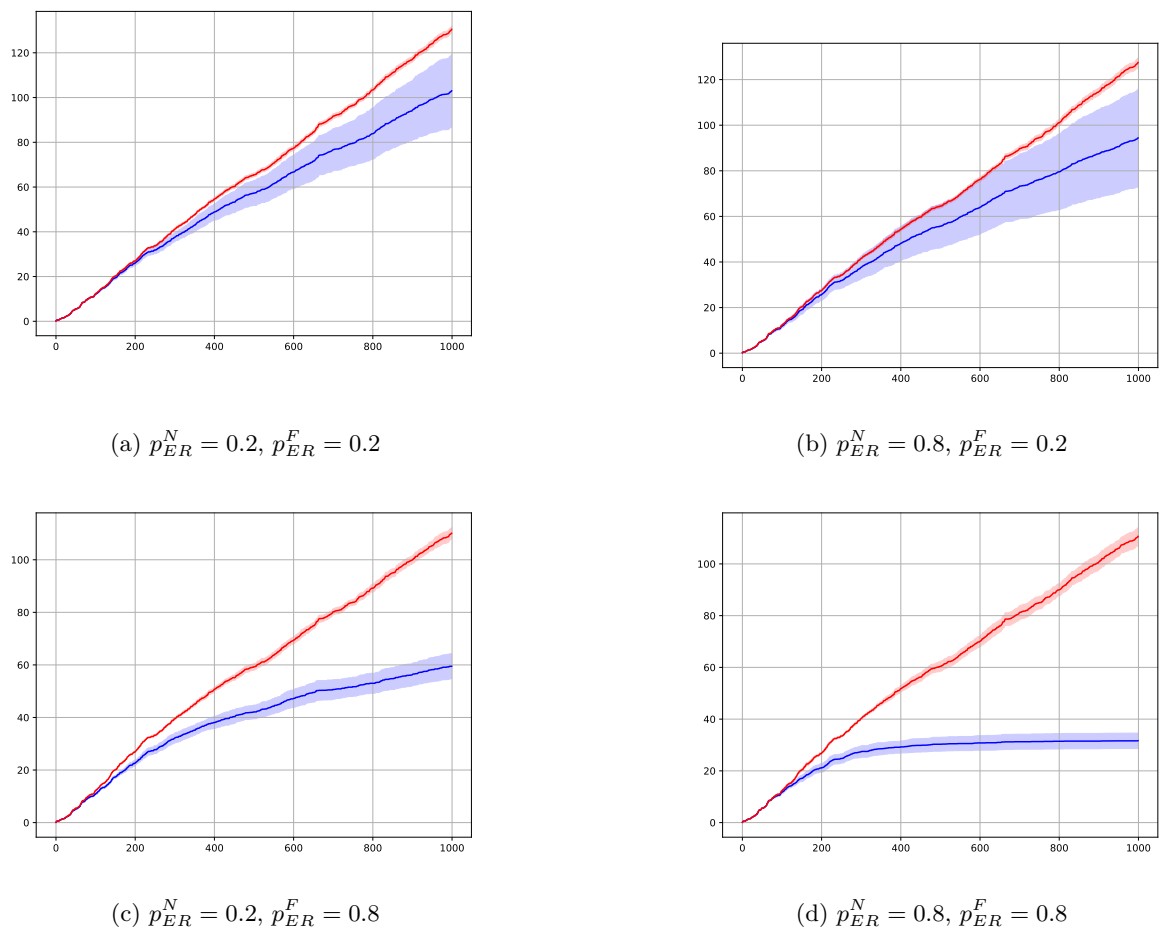

(a) $p_{ER}^N = 0.2$, $p_{ER}^F = 0.2$

(b) $p_{ER}^N = 0.8$, $p_{ER}^F = 0.2$

(c) $p_{ER}^N = 0.2$, $p_{ER}^F = 0.8$

(d) $p_{ER}^N = 0.8$, $p_{ER}^F = 0.8$

Figure 5: Average regret $R_T/Q$ against $T = 1000$ of rounds. Activation probability $q = 0.05$.

