_{\text{Bin}(q,T)}(m)\right)} = \frac{\sqrt{\alpha(F)}qT}{8}\frac{(qT)^2}{\sum_{m\in[T]}\left(m^{3/2}f_{\text{Bin}(q,T)}(m)\right)}$$

$$= \frac{\sqrt{\alpha(F)}qT}{8}\frac{(qT)^{3/2}}{\sum_{m\in[T]}\left(m^{3/2}f_{\text{Bin}(q,T)}(m)\right)}$$

where in the first equality we substituted the expected value of a binomial distribution of parameter $q$.

We now want to prove the existence of a constant $c > 0$ such that, for every $q > 0$ and every $T \geq \frac{1}{q^3}$

$$\frac{(qT)^{3/2}}{\sum_{m\in[T]}\left(m^{3/2}f_{\text{Bin}(q,T)}(m)\right)} \geq c, \quad \text{or equivalently} \quad \frac{\sum_{m\in[T]}\left(m^{3/2}f_{\text{Bin}(q,T)}(m)\right)}{(qT)^{3/2}} \leq c$$

We split the sum over $m \in [T]$ into two blocks, the first for $1 \leq m \leq c_2\lfloor qT\rfloor$ and the second for $c_2\lfloor qT\rfloor < m \leq T$ for a constant $c_2 = \left\lceil\frac{qT}{\lfloor qT\rfloor}\left(\frac{1}{q}\left(\sqrt{\frac{1}{2T}\ln\left(\frac{T^{3/2}}{c_1}\right)}-\frac{1}{T}\right)+1\right)\right\rceil$ and with $c_1 > 0$:

$$\frac{\sum_{m\in[T]}\left(m^{3/2}f_{\text{Bin}(q,T)}(m)\right)}{(qT)^{3/2}} = \frac{\sum_{m=1}^{c_2\lfloor qT\rfloor}\left(m^{3/2}f_{\text{Bin}(q,T)}(m)\right)}{(qT)^{3/2}} + \frac{\sum_{m>c_2\lfloor qT\rfloor}\left(m^{3/2}f_{\text{Bin}(q,T)}(m)\right)}{(qT)^{3/2}}. \tag{9}$$

The idea is to choose the split point $c_2\lfloor qT\rfloor$ so that we can upper bound the tail mass using Hoeffding's inequality. Hoeffding's inequality yields the simple bound $F_{\text{Bin}(q,T)}(m) \leq \exp\left(-2T\left(q-\frac{m}{T}\right)^2\right)$, and together with symmetry properties of the binomial distribution $1 - F_{\text{Bin}(q,T)}(m) = F_{\text{Bin}(1-q,T)}(T-m)$ we obtain a bound on the upper tail. This contribution compensates exactly the term $q^{3/2}$ left at the denominator for $T \geq 1/q$, leaving just the constant $c_1$:

$$\frac{\sum_{m>c_2\lfloor qT\rfloor}\left(m^{3/2}f_{\text{Bin}(q,T)}(m)\right)}{(qT)^{3/2}} \leq \frac{e^{-2T\left((1-q)-\frac{T-(c_2\lfloor qT\rfloor+1)}{T}\right)^2}T^{3/2}}{(qT)^{3/2}}$$

$$= \frac{e^{-2T\left(q\left(c_2\frac{\lfloor qT\rfloor}{qT}-1\right)+\frac{1}{T}\right)^2}}{q^{3/2}}$$

$$= \frac{c_1}{(qT)^{3/2}}$$

$$\leq c_1.$$

For the first term in Equation (9), we upper bound the lower tail simply by one:

$$\sum_{m=1}^{c_2\lfloor qT\rfloor}\left(m^{3/2}f_{\text{Bin}(q,T)}(m)\right) \leq c_2\lfloor qT\rfloor \cdot F_{\text{Bin}(q,T)}(m) \leq c_2\lfloor qT\rfloor$$

We conclude by proving that the first term in Equation (9) is bounded by a constant. If we take $m \leq c_2\lfloor qT\rfloor$ we obtain for $T \geq 2/q$ and $c_1 \geq 1$

$$\frac{\sum_{m=1}^{c_2\lfloor qT\rfloor}\left(m^{3/2}f_{\text{Bin}(q,T)}(m)\right)}{(qT)^{3/2}} \leq \frac{(c_2\lfloor qT\rfloor)^{3/2}}{(qT)^{3/2}}$$

$$\leq (c_2)^{3/2}$$