# OpenReview forum: "Cooperative Online Learning with Feedback Graphs"
_TMLR — Accepted by TMLR_

### Review · Reviewer_6GPK · 2024-01-28

**Summary Of Contributions:**

The paper considers a problem setup formulated as a common generalization of online learning with a feedback graph and corporative online earning with a communication graph.
This paper proposes an algorithm for this problem,
which achieves a regret upper bound that depends on the independence number of the strong product of the feedback graph and the communication graph.
This result is complemented by nearly tight lower bounds and numerical experiments.

**Audience:**

Yes

**Claims And Evidence:**

Yes

**Requested Changes:**

I do not believe that major changes are necessary, but I would appreciate it if you would consider addressing the concerns raised in the above weakness section.

**Strengths And Weaknesses:**

I believe that the research question in this paper,
how the structure of communication networks and feedback graphs affects the complexity of a problem,
is a topic of interest to the online learning theory community.
Also, from what I could see, no major errors were found in the theoretical analysis or numerical experiments.
In light of the above and the evaluation criteria of the TMLR, I support the acceptance of this paper.

On the other hand, I think this paper has the following weaknesses:
* The design and analysis of the algorithm are based on a straightforward combination of the existing studies in corporative online learning,
e.g., by Cesa-Bianchi et al. (2019),
with those in graph bandits,
e.g., by Alon et al. (2017).
The obtained upper bound does not look surprising.
* A gap remains between the upper bound and the lower bound.
In fact,
the lower bound applies only to the instances such that
$\alpha( N \boxtimes F) = \alpha(N) \alpha (F)$,
which means that the lower bound essentially depends on the value of $\alpha(N) \alpha(F)$ while
the upper bounds depends on $\alpha( N \boxtimes F)$.
I believe that the issue of which is tighter is the non-trivial and interesting question.
* In the contexts online learning with feedback graphs,
directed feedback graphs, graphs without feedback graph, and weakly observable graphs have been considered,
e.g., by Alon et al. (2017).
Such more general graphs are not addressed or even mentioned in this paper.
Not including them is not a problem in itself,
but it would be better to have an explanation as to why they are not included.
(For example, I would like to know if it is just for simplicity or for technical difficulty.)

---

> ### Author Response · Authors · 2024-03-09
>
> The paper is an attempt to answer the (seemingly natural) question of what is the right quantity in the bottom right corner of the table on page 2. We were honestly surprised to see the strong product of the two graphs appearing in the upper bound. And for this reason we agree that the question of determining the right scaling of the regret is non-trivial and intriguing. We conjecture that the correct scaling factor is that of the upper bound, as we now see how the strong product arises naturally from the mathematical structure of the model. The product of the independence numbers, instead, is relatively easy to force in a lower bound construction, and does not really exploit the interplay between the two graphs in the learning problem.
>
> The reviewer is right in asking for a discussion on the extension of our results to directed feedback graphs. We have added it to the conclusions. In short, we believe our results could be easily extended to the strongly observable case, where the scaling parameter remains the independence number. In the weakly observable case, both the scaling parameter and the rate change and it is not clear what one would obtain in our cooperative setting. We think this is an interesting question whose study, however, goes beyond the scope of this paper.

---

### Review · Reviewer_Mesd · 2024-02-22

**Summary Of Contributions:**

This work addresses the multi-agent online learning problem with feedback graphs, where agents can observe the losses associated with neighboring actions and receive information from nearby agents in the graph. The author proposes a novel algorithm that achieves regret dependent on the complexity of the graph. In several special situations, the results match previous findings, and an instance-based lower bound demonstrates that the upper bound is near-optimal. Additionally, experimental results support the efficiency of the proposed algorithm.

**Audience:**

Yes

**Broader Impact Concerns:**

I do not find any potential negative societal impact.

**Claims And Evidence:**

Yes

**Requested Changes:**

See weakness.

**Strengths And Weaknesses:**

Strengths:

1. The author proposes an algorithm for multi-agent online learning with general graph feedback that achieves regret dependent on the complexity of the graph. In several special situations, the results match previous findings, suggesting that the proposed algorithm can handle larger regions while providing similar guarantees.

2. The author also provides a lower bound, suggesting that the upper bound is near-optimal.

3. The paper is well-written and easy to follow.

4. Experimental results also support the efficiency of the algorithm.

Weakness:

1. It seems incorrect to claim that the results hold for an oblivious network. In the algorithm, the agent needs to know the network topology for agents within a distance no more than $n$. Otherwise, the agent cannot detect the generated time for a message.

2. The feedback message includes the draw distribution, which significantly increases the communication complexity.

3. For the cooperative learning process, the proposed algorithm requests a fixed participation probability $q(v)$, which seems restrictive. Recently, several works [1,2] have studied the cooperative learning process with asynchronous communication, allowing agents to decide whether to participate arbitrarily. It seems interesting to explore whether the current analysis can deal with this more general setting, and it would be beneficial if the author could include a discussion on this matter.

[1] Asynchronous upper confidence bound algorithms for federated linear bandits.

[2] A simple and provably efficient algorithm for asynchronous federated contextual linear bandits.

---

> ### Author Response · Authors · 2024-03-09
>
> - **It seems incorrect to claim that the results hold for an oblivious network.** We believe agents need only to know their neighbors. Messages are time-stamped (see item 3 in the box on page 4) via a global clock and dropped when older than the maximum delay. Hence, no agent can receive a message from an agent outside their communication radius. Yet, an agent may receive more than once the same message sent by an agent within their communication radius. To avoid updating twice, each agent can extract from the received messages the pairs $(t,v)$ (i.e., timestamp with agent index) and keep them stored for not longer than the maximum delay. We have added this observation to Section 3.
> - **The feedback message includes the draw distribution, which significantly increases the communication complexity.** True. Including the distribution in the messages is however essential to this type of analysis (see also N. Cesa-Bianchi, C. Gentile, and Y. Mansour, Delay and cooperation in nonstochastic bandits) and allows us to obtain previous results as special cases of ours.
> - **Deciding when to participate** We thank the reviewer for pointing out these additional related works which will be cited in the revision. Our results do rely on fixed participation probability. Previous works (e.g., N. Cesa-Bianchi, T. Cesari, C. Monteleoni, Cooperative online learning: keeping your neighbors updated) indeed proved a linear regret lower bound for arbitrary participation under oblivious network interface (Assumption 1). This fact is recalled on page 2.

---

### Review · Reviewer_bqdM · 2024-02-24

**Summary Of Contributions:**

This paper considers a problem of minimizing regret among  several cooperating agents. In the formal setup, we have a graph $F$ of $K$ possible “actions” and a graph $N$ of “agent connectivity”. In each round, each agent is activated with some probability, and then may take one action. It then receives feedback in the form of a loss for that  action, as well as the  “counterfactual” losses it would have received if it had chosen any action adjacent to the chosen action in $F$. Then it broadcasts this feedback as well as its distribution over actions. Broadcast messages reach nodes in the graph $N$ at distance $k$ from the broadcaster after $k$ rounds for an $k\le n$ for some given $n$ (and never reach nodes with $k>n$). The goal is to minimize the network regret, which is roughly the sum of all of the individual agent’s regrets with respect to a single shared comparator action.

This paper generalizes recent results that  consider the case of that $F$ is a clique (which corresponds to the standard “experts”  setting) and the case that $F$ has no edges (which corresponds to the standard  “multi-armed bandits” setting). The final regret is

$\sqrt{\alpha(N^n\boxtimes F) T}$

Where $\alpha$ represents the “independence number” (which is roughly like a sphere-packing / covering number generalized to graphs).

The algorithm is quite simple: each node essentially runs a version of the standard Exp3 MAB bandit algorithm, but uses a loss estimate that is improved by using information from other agents within distance $n$ in the connectivity graph. This improvement is counterbalanced by restricting the algorithm to only use loss estimates that are at least $n$ time steps in the past in order to compensate for potential delays in receiving feedback from other agents.

A lower bound is presented showing the tightess of this regret bound, along with some simulated experiments.

Overall, I think the quality of the results is significant enough to warrant acceptance. I do have two questions, and one more major concern however that I think the authors must address in order for the paper to be acceptable. I will enumerate them in the "strengths and weaknesses" section below.

**Audience:**

Yes

**Broader Impact Concerns:**

None.

**Claims And Evidence:**

No

**Requested Changes:**

Please address the weaknesses described above.

**Strengths And Weaknesses:**

# Strengths:
A simple algorithm that seems to optimally address a relatively clean problem. The analysis requires some algebra, but does not seem overly involved.

# Weaknesses/ Questions:

## First, a few more minor issues:

The communication model seems oddly limiting here and could do with more motivation: wouldn't we gain much more power if agents were allowed to broadcast something other than their own direct observations/action distributions? What if they could “pass on” messages from other agents? This doesn't seem that unrealistic, and yet it seems that it might significantly improve results. Is the $n$ message distance limitation supposed to factor-in some idea that after $n$ times being passed on a message will become corrupted and unreadable?  I am not an expert in distributed algorithms, but it seems to me that this issue could be more clearly explained.

On Page 8, regarding the argument that $B_s(i,v)$  and  $B_t(i,v)$  are independent: this point is  more subtle than is provided in the text. It appears that the key  idea is that because we only use \hat \ell_1, \dots,\hat\ell_{t-n}  at time t, this means that all actions taken using information from time steps for which there is no “in-flight” information. As written this  was confusing at first because naively it is easy to conceive of an algorithm that acts on information as soon as it is received. For such an algorithm, it might not hold that $B_s(i,v)$ and $B_t(i,v)$ are independent. So, I think some clarification in this part of the proof is warranted.

Page 9 just  before Jensen inequality, it would be clearer if the summation were not on top of the fraction: doesn’t make it  obvious that the square root  is part of the summation

## Last, a more significant issue:

I have a technical concern in Theorem 2, which  I hope the authors can clarify.  In particular, I do not understand how Lemma 2 is being applied here.

My understanding of the proof is that  the idea is to construct a graph for which the agents cannot communicate, and then argue that since the regret is the sum of the individual agents’ regrets, we apply Lemma 2 to each agent to obtain a regret that is (num agents) * $\sqrt{\alpha(F) Q T}$, where here “num algorithms” is really the “effective" number of agents after we’ve done some manipulation to ensure that no communication is possible.

Now, my concern is that Lemma 2 does NOT say that “for any graph $F$, there is  a sequence of losses such that the regret of  ANY agent is $\sqrt{\alpha(F) Q T}$”. Instead, it seems to say “For any graph $F$ and any agent, there exists a sequence of losses such that the regret  of the agent is at least $\sqrt{\alpha(F) Q T}$”.

As a result, I don’t follow how it is applied in the proof of theorem 2: it seems to me that although Lemma 2 can indeed provide losses that lower bound the regret of one of the agents, in doing so it may not be able to simultaneously lower bound the regret of all of the agents. Now, I’m willing to believe that this can be fixed, but it is not so obvious how and so needs some more detail.

For example, If I am allowed to have # agents > |K|, then I can just have one agent dedicated to playing each action and then clearly it is impossible to force all the agents to have low regret since one of them must have regret 0. Now, obviously this is not actually an algorithm that will have low regret overall, but it IS one that appears to break the argument currently used in the proof of Theorem 2.

Possibly I've misunderstood the way in which Lemma 2 is used, or missed some other subtlety. I have currently marked "No" on the "claims and evidence" question for this review, but if this concern can be alleviated, I will change to "yes".

---

> ### Author Response · Authors · 2024-03-09
>
> - **Wouldn't we gain much more power if agents were allowed to broadcast something other than their own direct observations/action distributions?** Yes, they could run complex distributed algorithms and, for example, acquire information about the global topology of the communication graph. Our work, however, follows a different approach: we study what can be achieved when agents just run simple online learning algorithms. As a consequence, our agents use messages only to transmit information which other agents can directly use to perform updates to their action distributions.
> - **What if they could “pass on” messages from other agents?**
> They actually do, see item 4 in the box on page 4. Each agent passes on any received message that are not older than the maximum delay. Messages older than that are automatically dropped.
> - **Is the message distance limitation supposed to factor-in some idea that after times being passed on a message will become corrupted and unreadable?** The limitation on the communication radius has been introduced to provide a way to control the message complexity (i.e., the number of messages traveling through the network).
> - **Independence of $B_s(i,v)$ and $B_t(i,v)$:** Fix any $i$, $v$, and $t>n$, and consider formula (3). The key observation is that $p_t(i,v)$ is determined when conditioning on $\mathcal H_{t-n}$ (or, formally, that $p_t(i,v)$ is $\mathcal H_{t-n}$-measurable). To see this, note that the probability $p_t(i,v)$ is a deterministic function of $w_t(i,v)$, which is in turn a deterministic function of  $\mathcal A_1, I_1, p_1, \dots \mathcal A_{t-n-1}, I_{t-n-1}, p_{t-n-1}$, which are all $\mathcal H_{t-n}$-measurable random variables (recall that $\mathcal H_{t-n}$ is the history up to and including time $t-n-1$). This shows that $p_t(i,v)$ is $\mathcal H_{t-n}$-measurable and, similarly, one gets that $p_s(i,v)$ is $\mathcal H_{t-n}$-measurable, for all $s\le t$.
> With this observation, and recalling that the $I_s$'s are drawn according to the $p_s$'s, we then simply note that for all $s\le t$, $B_s$ is a deterministic function of $\mathcal A_s$ and $I_s$, which, given $\mathcal H_{t-n}$, are both *independently* drawn at each time step according to fixed distributions.
> -  **Summation on top fraction:** fixed it.
> - **Lower Bound:**  We have been too concise in presenting the lower bound. We thank the reviewer for pointing out the need for a more detailed proof.
> The main observation, which we omitted to specify, is that the infimum in Theorem 2 is achieved when all agents run an instance of the same algorithm. Note that this rules out the reviewer's counterexample, where each agent is consistently pulling a different arm. The missing details have been added to the revision.

---

> > ### Comment · Reviewer_bqdM · 2024-03-15
> > **still confused by lower bound**
> >
> > In the lower bound on page 10, where did the max over actions go? You choose $\aleph^\star$ and $\ell^\star$. Do you not also need to choose $i^\star$ and then in the following long chain of inequalities the first one should have $\ell^\star_t(I^\star_t(v)) - \ell^\star_t(i^\star)$ instead of just $\ell^\star_t(I^\star_t(v))$?  This would be an issue because then in the choice of $v^\star$, the argmin would have already committed to $i^\star$.
> >
> > In fact, this concern highlights a more subtle issue that I do not follow: it is claimed that $R_T(q,\aleph', \ell^\star) \ge R_T(q,\aleph^\star, \ell^\star)$ due to the optimality of the *pair* $(\aleph^\star, \ell^\star)$. However, I'm not sure this is true. Consider for example the case of a simple non-network multi-arm bandit. Let $\aleph^\star,\ell^\star$ be an algorithm/loss sequence pair that achieves the minimax regret. It is still the case that there exists an algorithm $\aleph'$ such that $\aleph'$, when played using the loss sequence $\ell^\star$ will achieve lower regret than $\aleph^\star$: we can again simply choose an algorithm that just. always plays the best action in hindsight to achieve zero regret. This does not contradict the optimality of $\aleph^\star$ because there is a *different* loss sequence $\ell'$ for which $\aleph'$ would do poorly.
> >
> > Again, perhaps I misunderstood the argument, but so far I am not convinced.

---

> > > ### Author Response · Authors · 2024-04-03
> > >
> > > After a few more hours spent on the proof, we now agree with reviewer bqdM that our approach does not appear to lead to a lower bound for the case in which the algorithms run by the agents are not instances of the same online algorithm. As the reviewer correctly pointed out, the problem lies in exhibiting a single loss sequence that is worst-case simultaneously for all the instances. Luckily, our upper bound (Theorem 1) holds with the oblivious network interface, which implies that all agents are required to run an instance of the same algorithm with the same initialization. This is exactly the case that our lower bound construction can handle, as we show in the revised manuscript. This leaves open the intriguing question of whether our upper bound could be improved by using algorithms that do not run with an oblivious network interface while respecting the same communication constraints as before. In particular: no communication takes place when the active agents are elements of an independent set. We have added this question as an open problem in the revised manuscript.

---

### Decision · Action_Editor_RD2y · 2024-04-30

**Recommendation:** Accept with minor revision

**Comment:**

This work is worthy of publication in TMLR; my recommendation is for the paper to be accepted with minor revision. Regarding the minor revisions, the main thing is to look into the the issue of whether Theorem 2 holds only for deterministic algorithms (which seems likely) and update the theorem statement accordingly. Also, when you first mention your lower bound in the main text (around the top of page 2), I believe you need to add further qualifications (i.e., restrictions) for your result in light of Theorem 2 currently requiring all agents to use the same algorithm.

Below are some minor things (editorial in nature):
 - The table in the center of page 2 is currently an orphan. It should be numbered and I think also equipped with a caption.
 - In the paragraph immediately after the table in the center of page 2, change “is necessary to not incur in a linear regret” by removing “in” (stylistically, better yet, remove "in a").
 - Can you include punctuation in Assumption 1? I raise this point because the writing of the 3rd item makes it feel like some text got truncated ("and when" —> I was left wondering if there was more the authors meant to write here); having a period at the end will ensure the reader that the writing is as intended.
 - I suggest being consistent with punctuation at the end of equations. Sometimes, you do include a period when an equation is at the end of a sentence. Elsewhere, you do not. My recommendation is to always include punctuation when doing so is necessary for grammatical correctness.
 - I don't know if it is just me, and maybe none of the reviewers looked at Figure 2 much, but I have a very difficult time reading these 3D scatter plots. Please do a sanity check and see if there is a clearer way to present the information therein in a way that allows the reader to precisely read the plots. For example, for the red and blue dots closest to (0.8, 0.2) in each of the 3 figures, it's very difficult to read the information in the plot. Given the few data points you have here, maybe there is a non-3D plot way to present the information.

**Audience:**

This work is of interesting to the online learning (theory) community.

**Claims And Evidence:**

There was a concern regarding the proof of the lower bound (Theorem 2). This has been addressed by the authors by weakening the theorem statement; the current version of Theorem 2 now restricts its scope to the case where all agents use the same algorithm and the network interface is oblivious. The original concern was raised by Reviewer bqdM, and this reviewer appears to be satisfied provided that the authors mention that this lower bound result is only known to hold for deterministic algorithms. In more detail, the reviewer writes
>"If possible, I would request that the authors be required to specify that the present lower bound holds only for deterministic algorithms: I believe that a randomized algorithm could randomize over its initialization state and cause some issues in the lower bound. As currently stated and argued, the lower bound contains no expectations or other reference to randomization so this should be fairly uncontroversial."


I ask the authors to please make this change unless they are able to see how the guarantee in fact holds for randomized algorithms and, if so, can adjust their proof accordingly.

There are no other outstanding concerns regarding correctness of results, so the claims are supported by evidence (proofs).

---

> ### Author Response · Authors · 2024-06-04
>
> Dear Action Editor and reviewers,
>
> Thank you very much for your valuable feedback and for accepting our work. We implemented all requested changes and will submit the final version shortly. In particular, we mention here that the lower bound does hold for randomized algorithms. The confusion was perhaps due to the fact that in some works, sometimes authors use $R_T$ to define the _realized_ regret, while we introduced it directly as the _expected_ regret. We stressed this more in the paper and added extra explanations in the proof of the lower bound to eliminate any possible misunderstandings.
>
> Best,
>
> The authors